# Data-Driven Priors for Uncertainty-Aware Deterioration Risk Prediction with Multimodal Data

## Abstract

Safe predictions are a crucial requirement for integrating predictive models into clinical decision support systems. One approach for ensuring trustworthiness is to enable models' ability to express their uncertainty about individual predictions. However, current machine learning models frequently lack reliable uncertainty estimation, hindering real-world deployment. This is further observed in multimodal settings, where the goal is to enable effective information fusion. In this work, we propose `MedCertAIn`, a predictive uncertainty framework that leverages multimodal clinical data for in-hospital risk prediction to improve model performance and reliability. We design data-driven priors over neural network parameters using a hybrid strategy that considers cross-modal similarity in self-supervised latent representations and modality-specific data corruptions. We train and evaluate the models with such priors using clinical time-series and chest X-ray images from the publicly-available datasets MIMIC-IV and MIMIC-CXR. Our results show that `MedCertAIn` significantly improves predictive performance and uncertainty quantification compared to state-of-the-art deterministic baselines and alternative Bayesian methods. These findings highlight the promise of data-driven priors in advancing robust, uncertainty-aware AI tools for high-stakes clinical applications.

## 1 Introduction

Trustworthy implementation of machine learning models in healthcare requires robust uncertainty measurements (Begoli et al., 2019; Gruber et al., 2023), considering the safety-critical nature of clinical practice. Uncertainty Quantification (UQ) approaches provide an essential layer of reliability in addition to accurate machine learning models for improved and safer decision-making (Nemani et al., 2023). Uncertainty in machine learning models can be due to model parameters, noise and bias of the calibration data, or deployment of the model in an out-of-distribution scenario (Miller et al., 2014). All of these are inherent characteristics of any real-world clinical scenario. Artificial Intelligence (AI) systems that reliably provide uncertainty metrics can enhance the confidence of healthcare professionals when using these systems and, at the same time, improve the effectiveness of the predictions (Seoni et al., 2023) .

Nonetheless, the study of UQ within the area of machine learning for healthcare has been limited (Kompa et al., 2021; Lechuga et al., 2025). A comprehensive understanding of the uncertainty present in healthcare applications and how to address it is still fragmented (Han et al., 2011), which may be due to the limited underlying theory on how to best adapt predictive uncertainty for clinical tasks (Begoli et al., 2019). Other reasons include the complexity of scaling UQ in real-time clinical systems, limited empirical evaluation of different methods due to the lack of well-constructed priors by medical experts (Kurz et al., 2022), and the high prevalence of data shifts in real-world clinical applications. All of these can negatively affect predictive performance (Ovadia et al., 2019; Xia et al., 2022), further emphasizing the need for better measurements of uncertainty in predictive models.

In addition, developing machine learning systems that are suitable for healthcare applications require the use of different data modalities simultaneously, to reflect the multimodal nature of the human decision-making process (Saab et al., 2024). However, existing work on UQ in healthcare has been mainly studied in the

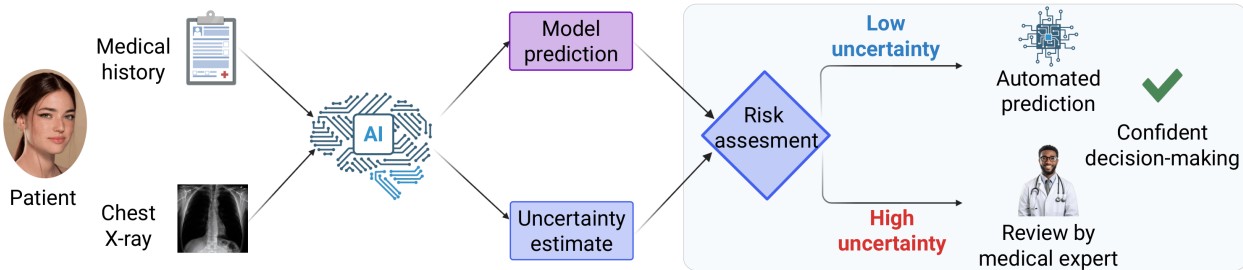

Figure 1: **Uncertainty-Aware Clinical Decision-making.** Patient multimodal data is processed by our uncertainty-aware model to produce a prediction and an uncertainty estimate. Clinicians accept confident predictions or defer uncertain cases for further review, improving workflow efficiency and supporting reliable decision-making.

unimodal setting, with a particular focus on medical imaging applications (Gawlikowski et al., 2021). Hence, effective quantification of predictive uncertainty in the context of multimodal clinical problems remains a challenging and unsolved task (Tran et al., 2022).

To address these gaps, we introduce `MedCertAIn`, a multimodal uncertainty-aware framework for deterioration risk prediction that explicitly recognizes and defers low-confidence cases for human review, which is considered a key requirement for real-world clinical decision support (Figure 1). Our method combines Bayesian learning with automated, label-free context-prior construction to enable selective prediction without manual annotation or domain-specific expert input. In this work, we evaluate this framework across multiple clinically relevant mortality prediction horizons using paired EHR–CXR data. We summarize our key contributions as follows:

1. We propose `MedCertAIn`, a multimodal uncertainty-aware framework that combines Bayesian learning and variational inference to augment existing architectures with principled uncertainty estimates and improved reliability.

2. We design a flexible label-free pipeline for automatically constructing multimodal priors by considering cross-modal similarity in self-supervised latent representations and modality-specific input perturbations, eliminating the need for expert-annotated high-uncertainty context sets.

3. We train and evaluate our framework on publicly available multimodal data comprising clinical time-series and chest X-ray images from MIMIC-IV (Johnson et al., 2021) and MIMIC-CXR (Johnson et al., 2019). Our findings demonstrate improved predictive performance and reliability.

4. We implement our framework in `JAX` and make the code publicly available to promote future research in uncertainty-aware multimodal clinical modeling: anonymous repository.

The paper is organized as follows: Section 2 provides an overview of related work on multimodal learning and UQ in healthcare, Section 3 presents the proposed `MedCertAIn` framework and training methodology, Section 4 describes the experimental setting, Section 5 presents the results and empirical findings, and Section 6 discusses the implications, limitations, and future directions.

## 2 Related Work

### 2.1 Multimodal Learning in Healthcare

Multimodal learning approaches seek to leverage complementary information from different data modalities to enhance the models' predictive capabilities (Hayat et al., 2022). In healthcare, this paradigm is particularly relevant due to the heterogeneous nature of clinical data, which often includes structured electronic health records, medical imaging, physiological signals, and free-text reports. Many approaches for combining information across modalities exist, with multimodal fusion being the most widely adopted strategy (Huang et al., 2020; Hayat et al., 2022).

Early work primarily focused on integrating multiple imaging modalities or views for tasks such as segmentation and disease characterization, particularly in neuroimaging (Zhang et al., 2020; Calhoun & Sui, 2016). More recent studies have extended multimodal learning to combine fundamentally different data types, such as imaging with time-series or tabular clinical data, demonstrating improved performance in prognostic and risk prediction tasks (Muhammad et al., 2021). This trend is exemplified by multimodal systems developed for patient deterioration and outcome prediction, including COVID-19 prognosis using combined imaging and clinical variables (Shamout et al., 2021; Jiao et al., 2021). While multimodal integration enables richer patient representations, it also introduces additional sources of uncertainty arising from heterogeneity across modalities, imperfect alignment in time and semantics, and potential disagreement between modality-specific signals (Azizmalayeri et al., 2025). These challenges are inherent of real-world scenarios motivating the need for modeling approaches that explicitly account for uncertainty in multimodal settings (Kurz et al., 2022).

## 2.2 Uncertainty Quantification in Medical Applications

The safety-critical nature of clinical decision-making has motivated growing interest in UQ for machine learning models in healthcare (Gawlikowski et al., 2021; Seoni et al., 2023). Most existing work on UQ in medical applications has focused on unimodal settings, particularly medical imaging, where uncertainty estimates have been used to support tasks such as brain tumor segmentation (Jungo et al., 2018), skin lesion classification (DeVries & Taylor, 2018), and diabetic retinopathy detection (Filos et al., 2019; Band et al., 2021; Nado et al., 2022). These studies demonstrate the potential of UQ to improve interpretability and model trust by distinguishing confident predictions from ambiguous cases.

However, uncertainty in real-world healthcare systems arises from multiple sources, including data noise, dataset shift, model misspecification, and deployment in out-of-distribution clinical environments (Lechuga et al., 2025). Existing UQ approaches rarely account for these factors jointly, nor do they explicitly address the challenges introduced by multimodal clinical data, particularly those designed to support selective prediction and deferral in deployment settings, where uncertainty may stem from modality disagreement or missing and corrupted inputs (Liang et al., 2023).

## 2.3 Stochastic Neural Networks and Variational Inference

Stochastic neural networks extend deterministic neural networks by treating their parameters as random variables rather than fixed point estimates. This probabilistic formulation enables the model to capture epistemic uncertainty and provides a principled foundation for Bayesian inference over network weights. Hence, for a given neural network $f(\cdot)$, a stochastic version $f(\cdot\,;\Theta)$ is defined in terms of stochastic parameters $\Theta$. For an observation model $p_{Y|X,\Theta}$ and a prior distribution over parameters $p_\Theta$, Bayesian inference provides a principled framework for modeling predictive uncertainty by inferring the posterior distribution over parameters given the observed data, $p_{\Theta|\mathcal{D}}$ (MacKay, 1992; Neal, 1996). However, since neural networks are non-linear in their parameters, exact inference over the stochastic network parameters is analytically intractable. Variational inference is an approach that seeks to avoid this intractability by framing posterior inference as finding an approximation $q_\Theta$ to the posterior $p_{\Theta|\mathcal{D}}$ via the variational optimization problem:

$$\min_{q_\Theta \in \mathcal{Q}_\Theta} \mathbb{D}_{\mathrm{KL}}(q_\Theta \,\|\, p_{\Theta|\mathcal{D}}) \iff \max_{q_\Theta \in \mathcal{Q}_\Theta} \mathcal{F}(q_\Theta)$$

where $\mathcal{F}(q_\Theta)$ is the variational objective:

$$\mathcal{F}(q_\Theta) \doteq \mathbb{E}_{q_\Theta}[\log p(y_\mathcal{D} \,|\, x_\mathcal{D}, \Theta)] - \mathbb{D}_{\mathrm{KL}}(q_\Theta \,\|\, p_\Theta),$$

$\mathcal{Q}_\Theta$ is a variational family of distributions (Wainwright & Jordan, 2008), $\mathbb{D}_{\mathrm{KL}}$ is the Kullback-Leibler divergence (Kullback & Leibler, 1951), and $(x_\mathcal{D}, y_\mathcal{D})$ are the training data. One particularly simple type of variational inference is Gaussian mean-field variational inference (Blundell et al., 2015; Graves, 2011), where the posterior distribution over network parameters is approximated by a Gaussian distribution with a diagonal covariance matrix. This method enables stochastic optimization and can be scaled to large neural networks (Hoffman et al., 2013). However, Gaussian mean-field variational inference has been shown to underperform with deterministic neural networks when uninformative, standard Gaussian priors are used (Ovadia et al., 2019; Rudner et al., 2022).

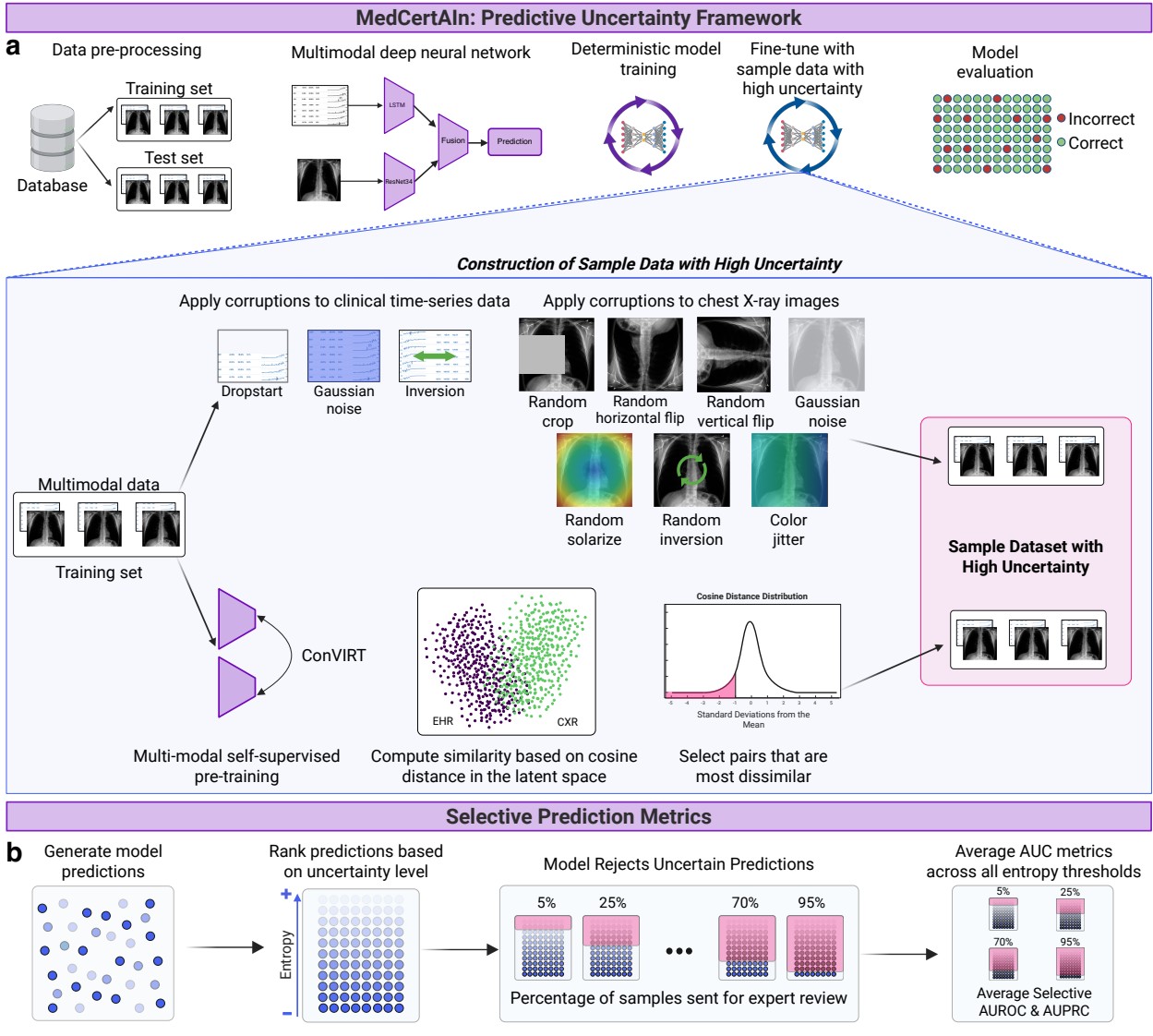

Figure 2: **MedCertAIn: A Predictive Uncertainty Framework a)** We construct paired multimodal samples for training/evaluation and generate high-uncertainty context data. From the training set, we create corrupted pairs via noise and transformations, and additionally pre-train a self-supervised model to select low-similarity pairs in latent space, reflecting high cross-modal uncertainty. We combine these context sets and fine-tune the model using Bayesian learning, evaluating with standard ROC and selective prediction metrics. **b) Selective Prediction Metrics:** We rank predictions using Shannon entropy and evaluate performance on retained subsets across coverage thresholds (0–100%). Averaging across thresholds yields selective AUROC/AUPRC, enabling clinicians to choose a deferral point where the model is least reliable.

# 3 Method

`MedCertAIn` is a multimodal uncertainty-aware learning framework designed to explicitly refer low-confidence cases in safety-critical clinical settings for expert review. Our contribution is an uncertainty-aware training objective that augments standard multimodal prediction with principled uncertainty estimation through context-prior construction and loss regularization. Rather than requiring architectural changes to the fusion network, `MedCertAIn` operates at the level of the optimization objective, enabling reliable uncertainty-aware selective prediction.

### 3.1 Deriving the Optimization Objective

We consider a supervised multimodal fusion task on a dataset $\mathcal{D} \doteq \{(x_n^{ehr}, x_n^{cxr}, y_n)\}_{n=1}^N = (X_{\mathcal{D}}^{ehr}, X_{\mathcal{D}}^{cxr}, Y_{\mathcal{D}})$, where each sample consists of paired electronic health record time-series and chest X-ray images. As illustrated in Figure 2.a, the two modalities are processed by modality-specific encoders $\Phi^{ehr}$ and $\Phi^{cxr}$, respectively. The resulting feature representations are concatenated and passed to a classifier $g(\cdot)$ to produce a fusion prediction $\hat{y}$, which is optimized using supervised learning with respect to the ground truth labels $y \in \{0, 1\}$.

To mitigate the limitations of variational inference under non-informative priors, we use information from our two input modalities to construct data-driven priors that favor parameter values inducing reliable uncertainty-aware predictions (Rudner et al., 2024b). We consider a standard parametric observation model $p_{Y|X,\Theta}(y \,|\, x, \theta; f)$, and let the mapping $f$ be defined by $f(\cdot; \theta) \doteq h(\cdot; \theta_h)\theta_L$, where $h(\cdot; \theta_h)$ is the post-activation output of the penultimate layer, $\Theta_L$ is the set of stochastic final-layer parameters, $\Theta_h$ is the set of stochastic non-final-layer parameters, and $\Theta \doteq \{\Theta_h, \Theta_L\}$ is the full set of stochastic parameters. In preparation for constructing a data-driven prior, we denote a batch of informative context inputs with corresponding context labels by $x_c = \{x_1, \ldots, x_M\}, y_c = \{y_1, \ldots, y_M\}$, respectively, and let $p_{X_c, Y_c}$ be a joint distribution over context batches. The context inputs are chosen to represent a priori high-uncertainty regions, such as corrupted or cross-modally ambiguous samples which can help guide further clinical decision-making.

To construct a uncertainty-aware data-driven prior, we consider a Bernoulli random variable $\check{Z}$, denoting whether a given set of neural network parameters induces predictions that exhibit some desired property (e.g., high uncertainty on a set of evaluation points). Furthermore, we define a *context observation model* $\check{p}_{\check{Z}|\Theta}(\check{z} \,|\, \theta; f, p_{X_c, Y_c})$—which denotes the likelihood of observing an auxiliary outcome $\check{z}$ under $\check{p}_{\check{Z}|\Theta}$, given $\theta$, and $p_{X_c, Y_c}$—and specify a *base prior* over the model parameters, $p_\Theta(\theta)$. Here, the event $\check{z} = 1$ corresponds to the parameter setting inducing the desired uncertainty behavior on the context points. Using the context data and the above context observation model, we can define a data-driven prior $p(\theta \,|\, \check{z}; p_{X_c, Y_c})$, conditioned on the desired event occurring.

Unfortunately, since the full distribution $p(\theta \,|\, \check{z}; p_{X_c, Y_c})$ is intractable for all but the simplest context observation models, we cannot use this prior directly. To incorporate it into training nevertheless, in Appendix C.1, we show how to obtain a tractable optimization objective that does not require computing the full data-driven prior density and instead only requires specifying an auxiliary context cost function, $c(X_c, Y_c, \Theta)$. This cost function defines the context observation model and is chosen to penalize parameter values $\theta$ that induce predictions for which the expected cost under $p_{X_c, Y_c}$ is high.

For training our stochastic model, we therefore extend the variational objective with an uncertainty regularization term that makes use of the set of context points:

$$\mathcal{F}(q_\Theta) \doteq \underbrace{\mathbb{E}_{q_\Theta}[\log p(y_{\mathcal{D}} \,|\, x_{\mathcal{D}}, \Theta; f)]}_{\text{Expected log-likelihood}} - \underbrace{\mathbb{D}_{\text{KL}}(q_\Theta \,\|\, p_\Theta)}_{\text{KL regularization}} - \underbrace{\mathbb{E}_{q_\Theta}[\mathbb{E}_{p_{X_c, Y_c}}[c(X_c, Y_c, \Theta)]]}_{\text{Uncertainty regularization}}.$$

Letting $p_{Y_c|X_c}(y_c \,|\, x_c) = \delta(\mathbf{0} - y_c)$ to encourage high uncertainty in the predictions on the set of context points, where $\delta(\cdot)$ is the Dirac delta function, we obtain the simplified objective:

$$\mathcal{F}(q_\Theta) \doteq \underbrace{\mathbb{E}_{q_\Theta}[\log p(y_{\mathcal{D}} \,|\, x_{\mathcal{D}}, \Theta; f)]}_{\text{Expected log-likelihood}} - \underbrace{\mathbb{D}_{\text{KL}}(q_\Theta \,\|\, p_\Theta)}_{\text{KL regularization}} - \underbrace{\mathbb{E}_{q_\Theta}[\mathbb{E}_{p_{X_c}}[c(X_c, \mathbf{0}, \Theta)]]}_{\text{Uncertainty regularization}}.$$

In practice, each training step performs a forward pass on the combined training and context sets. We optimize the expected log-likelihood using binary cross-entropy on the supervised training labels, and compute the KL and uncertainty regularization terms using Monte Carlo estimates under $q_\Theta$. Gradients are obtained via reparameterization of gradients for mean-field variational inference (Blundell et al., 2015). The complete derivation of the tractable objective under the proposed uncertainty-aware prior is provided in Appendix C.

### 3.2 Constructing the Context Set in A Priori High-Uncertainty Regions

We define a context set $(X_c, Y_c)$ drawn from a distribution $p_{X_c, Y_c}$, representing samples in a priori high-uncertainty regions (i.e., distributionally shifted or ambiguous points). From the multimodal training set

$(X_{\mathcal{D}}^{ehr}, X_{\mathcal{D}}^{cxr})$, we construct a context set $(X_{\mathcal{C}}^{ehr}, X_{\mathcal{C}}^{cxr})$ using two label-free strategies which enable data-driven prior construction with minimal human intervention: (1) applying controlled corruptions to both modalities, and (2) selecting samples with low cross-modal similarity in a self-supervised latent space. The latter is motivated by the assumption that modality mismatch corresponds to higher predictive uncertainty.

### 3.2.1 Modality-specific Data Perturbations

We induce a controlled distribution shift by applying data corruptions to the original training set. For EHR time-series, we construct $X_{\mathcal{C}}^{ehr}$ by: (i) dropping an initial segment of the sequence up to a threshold, (ii) adding Gaussian noise, and (iii) reversing the time dimension. For chest X-rays, we construct $X_{\mathcal{C}}^{cxr}$ using seven perturbations: random crop, horizontal/vertical flip, Gaussian blur, solarize, invert, and color jitter. Corruption magnitudes are fixed during training using commonly adopted augmentation settings.

### 3.2.2 Cross-Modal Similarity in Multimodal Latent Space

Extending previous work (Lopez et al., 2023), we also construct a context set by selecting samples whose modality representations are dissimilar in a self-supervised latent space. We obtain per-modality representations $(\Phi^{ehr}, \Phi^{cxr})$ using a ConVIRT model trained with the infoNCE loss (Zhang et al., 2022; Hayat et al., 2022). We compute cosine similarity scores and select samples falling in the left tail of the similarity distribution.

**Inter-modal similarity.** For each training sample, we compute the cross-modal cosine similarity for each training sample and define the selection threshold $t_1$:

$$t_1 = \bar{\alpha} - v \cdot \sigma_\alpha, \qquad \alpha_i = \cos(\Phi_i^{ehr}, \Phi_i^{cxr}), \qquad \bar{\alpha} = \frac{1}{n}\sum_{i=1}^n \alpha_i, \qquad \sigma_\alpha = \text{std}\{\alpha_i\}_{i=1}^n, \qquad v \in [1, 2].$$

We define the selected index set as $\mathcal{C}_1 = \{i \in \{1, \ldots, n\} : \alpha_i < t_1\}$, and the corresponding context sets:

$$X_{\mathcal{C}_1}^{ehr} = \{X_i^{ehr} : i \in \mathcal{C}_1\}, \qquad X_{\mathcal{C}_1}^{cxr} = \{X_i^{cxr} : i \in \mathcal{C}_1\}.$$

**Inter- and intra-modal similarity.** We additionally incorporate intra-modal similarity between each training sample and the mean latent representation vector for each modality:

$$\beta_i = \cos\left(\Phi_i^{ehr}, \overline{\Phi}^{ehr}\right), \qquad \gamma_i = \cos\left(\Phi_i^{cxr}, \overline{\Phi}^{cxr}\right), \qquad \overline{\Phi}^{ehr} = \frac{1}{n}\sum_{i=1}^n \Phi_i^{ehr}, \qquad \overline{\Phi}^{cxr} = \frac{1}{n}\sum_{i=1}^n \Phi_i^{cxr}.$$

We aggregate the similarities by simple averaging and define the selection threshold $t_2$:

$$t_2 = \bar{\delta} - v\sigma_\delta, \qquad \delta_i = \frac{\alpha_i + \beta_i + \gamma_i}{3}, \qquad \bar{\delta} = \frac{1}{n}\sum_{i=1}^n \delta_i, \qquad \sigma_\delta = \text{std}\{\delta_i\}_{i=1}^n.$$

We define the selected index set as $\mathcal{C}_2 = \{i \in \{1, \ldots, n\} : \delta_i < t_2\}$, and the corresponding context sets:

$$X_{\mathcal{C}_2}^{ehr} = \{X_i^{ehr} : i \in \mathcal{C}_2\}, \qquad X_{\mathcal{C}_2}^{cxr} = \{X_i^{cxr} : i \in \mathcal{C}_2\}.$$

Finally, for training `MedCertAIn`, the high-uncertainty sample set is constructed by combining the corrupted training examples with either the inter-modal samples $\mathcal{C}_1$ or the inter- and intra-modal samples $\mathcal{C}_2$ selected using self-supervised latent-space similarity.

## 3.3 Uncertainty Measurement

To quantify predictive uncertainty, we compute uncertainty scores individually for each test sample based on the predictive distribution of the model. For a given input $x$, the stochastic network induces a predictive distribution $p(y \mid x)$, obtained via Monte Carlo sampling from the variational posterior. We quantify

uncertainty using the Shannon entropy of the predictive distribution, where higher entropy indicates greater predictive uncertainty:

$$\mathcal{H}(p(y \mid x)) = -\sum_{c \in \{0,1\}} p(y = c \mid x) \log p(y = c \mid x).$$

Shannon entropy provides a simple and widely used proxy for predictive uncertainty in Bayesian neural networks (Rudner et al., 2023a;b). In our setting, entropy scores are computed per data point and used to rank predictions for selective prediction which evaluates the models skill at detecting points that the model would recommend for further review, enabling uncertainty-aware clinical decision support.

## 4 Experimental Setting

### 4.1 Dataset

To train and evaluate our framework, we used two publicly available modalities: EHR time-series data from MIMIC-IV (Johnson et al., 2021) and chest X-ray images from MIMIC-CXR (Johnson et al., 2019). MIMIC-IV contains more than 60,000 ICU stays collected between 2001 and 2019. We only used complete paired samples, such that each example includes both modalities with no missing data. We focus on in-hospital mortality prediction, a clinically relevant task for ICU decision support systems (Sadeghi et al., 2018; Awad et al., 2017), to support clinical decision-making and resource allocation by identifying patients at higher risk of deterioration, enabling timely intervention (Rajpurkar et al., 2022). Mortality prediction is a binary classification task that estimates in-hospital mortality

Table 1: **Dataset Summary**: Summary of multimodal data samples used during training for the in-hospital mortality task. Only our proposed method `MedCertAIn` makes use of the high-uncertainty set (5,347 samples) which is constructed by combining corrupted training examples with 862 additional samples selected via cross-modal similarity in the self-supervised latent space.

| Model
Data split | Baseline Models | MedCertAIn |
|---|---|---|
| Training | 4,485 | 4,485 |
| Validation | 488 | 488 |
| Test | 1,242 | 1,242 |
| High Uncertainty Set | – | 5,347 |

after 48 hours in the ICU (Hayat et al., 2022). We excluded ICU stays shorter than 48 hours, resulting in 6,215 paired patient samples. We used an 80/20 train-test split and performed five-fold cross-validation on the training set with a 70/10 train-validation ratio. All splits are disjoint at the patient level. We follow Hayat et al. (2022) to pre-process and align the clinical time-series data and chest X-ray images for this task. Table 1 summarizes the multimodal sample sizes across training, validation, test and context set splits used for `MedCertAIn`.

### 4.2 Multimodal Backbone Neural Network

We adopt MedFuse (Hayat et al., 2022) as the backbone neural network architecture, a simple and robust fusion module for multimodal clinical prediction using paired EHR time-series and imaging data. MedFuse follows an early-fusion design, where modality-specific feature extractors encode each input before combining their representations for downstream prediction. Given a paired multimodal sample $(x^{ehr}, x^{cxr}) \in \mathcal{D}$, $\Phi_{\text{ehr}}$ encodes EHR time-series using a two-layer long short-term memory (LSTM) network (Hochreiter & Schmidhuber, 1997), which captures temporal dependencies in sequential clinical measurements. Chest X-ray images are encoded using a ResNet-34 convolutional neural network $\Phi_{\text{cxr}}$ (He et al., 2015), which extracts high-level spatial and semantic features from medical images. The resulting modality-specific representations are concatenated and passed to a classifier $g(\cdot)$ with a sigmoid activation to produce the predicted mortality risk $\hat{y}$.

### 4.3 Model Baselines

We compare `MedCertAIn` against three primary baselines. First, we use MedFuse (Hayat et al., 2022), our deterministic multimodal backbone, which serves as the architectural baseline without uncertainty-

aware training. Second, we evaluate DrFuse (Yao et al., 2024), a transformer-based multimodal model that disentangles shared and modality-specific representations and dynamically weights EHR and imaging inputs. We adapt DrFuse from clinical condition classification to the in-hospital mortality prediction task. Finally, we include a stochastic baseline based on Gaussian mean-field variational inference with an uninformative prior (Blundell et al., 2015; Graves, 2011), isolating the effect of Bayesian learning without informative, data-driven priors. All baselines are evaluated under identical data splits and experimental protocols to ensure fair comparison.

## 4.4 Evaluation Metrics

We evaluate test performance using AUROC and AUPRC. To assess predictive uncertainty, we additionally compute selective prediction metrics. Selective prediction introduces a "reject option" $\perp$ via a selection function $s : \mathcal{X} \to \mathbb{R}$ that determines whether to output a prediction for an input $x \in \mathcal{X}$ (El-Yaniv et al., 2010). Given a rejection threshold $\tau$ and using entropy as the selection score, the prediction rule is:

$$(p(y \mid \cdot, \theta; f), s)(x) = \begin{cases} p(y \mid x, \theta; f), & \text{if } s \leq \tau \\ \perp, & \text{otherwise.} \end{cases}$$

We compute AUROC and AUPRC over rejection thresholds $\tau = 0\%, \ldots, 99\%$ and report the average across thresholds as selective AUROC and selective AUPRC. Selective prediction metrics quantify predictive performance together with a model's uncertainty, providing an intuitive measure of reliability. In practice, they offer a simple mechanism to flag uncertain predictions for clinical review, improving interpretability for clinicians while ensuring that only low-confidence cases are deferred, optimizing clinical resources.

## 4.5 Hyperparameter Tuning and Model Selection

### 4.5.1 Deterministic models

For each task, we run 50 hyperparameter configurations, each evaluated over five runs with different random seeds (250 runs total). For every run, we randomly sample train/validation splits and initialize model weights. We sample the learning rate uniformly from $[10^{-5}, 10^{-2}]$, select the regularizer scale from $\{0, 0.1, 1, 10, 100\}$, and the number of training epochs from $\{5, 10, 15, 20, 30\}$. We use a fixed batch size of 16 and cosine decay with $\alpha = 0$. For each run, we save the checkpoint from the final training epoch and select the configuration with the highest average validation AUROC across the five seeds. Using the selected configuration, we retrain on the combined training and validation data and report test performance averaged over five random initializations. We compute standard errors across these five final runs.

### 4.5.2 Stochastic models

We initialize stochastic models from the five best deterministic checkpoints and perform hyperparameter tuning using the same protocol (50 configurations $\times$ 5 seeds; 250 runs per task). The learning rate is sampled one order of magnitude above and below the best deterministic learning rate, and training epochs are sampled from $\{5, 10, 15, 20, 25, 30\}$. We use batch size 16, cosine decay with $\alpha = 0$, and sample the context batch size from $\{16, 32\}$ due to the additional context data used for Bayesian learning. As stochastic models include mean and variance parameters and require forward passes over sampled context points, fine-tuning is more computationally expensive than the deterministic setting. Table D.11 summarizes the stochastic search space. With the optimal configuration, we train on the full training plus validation data and evaluate five stochastic models with different random initializations (5 runs). This procedure is repeated for each stochastic variant trained with different high-uncertainty context sets.

All models are trained with Adam and evaluated on the test set using means and standard errors over five seeds. Experiments were run on NVIDIA A100 and V100 80GB Tensor Core GPUs.

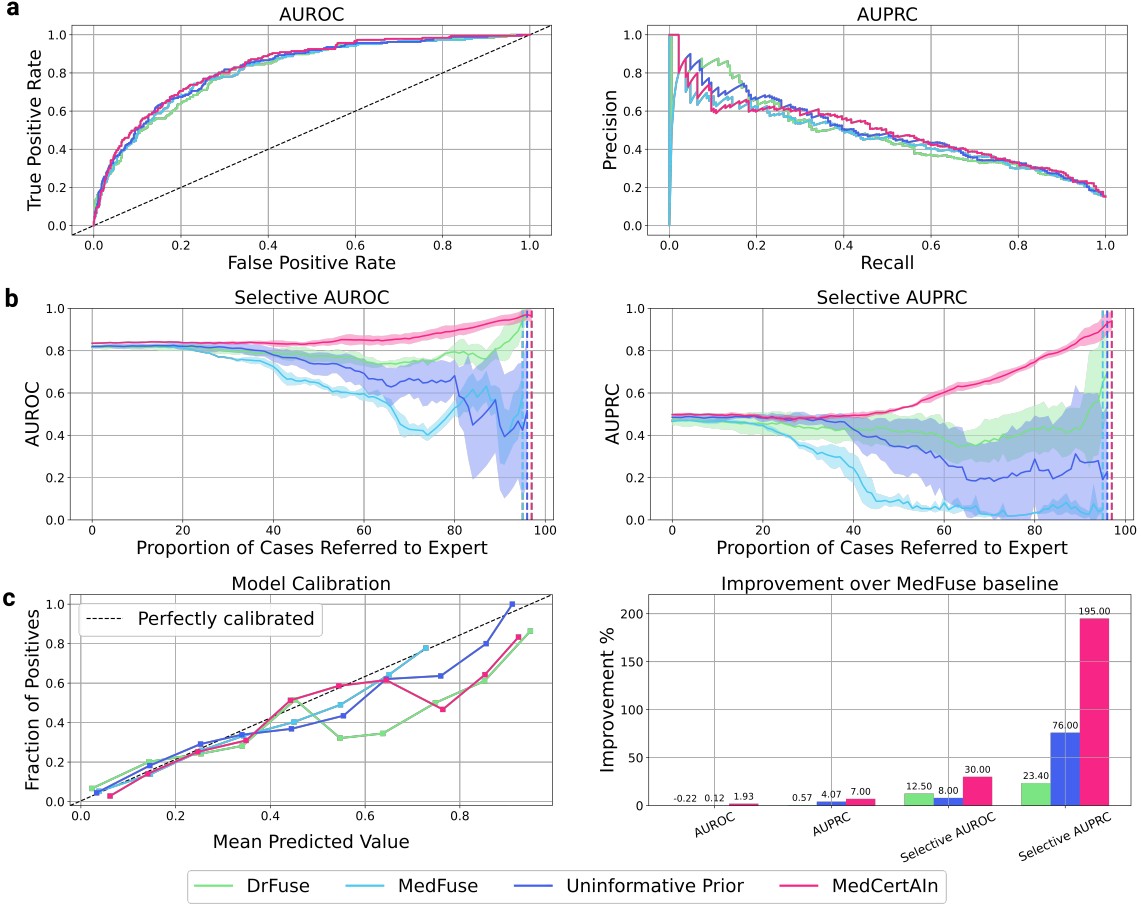

Figure 3: **MedCertAIn Achieves State-Of-The-Art Performance and Reliable Selective Prediction. a)** In standard metrics, the differences across baselines are very small for predictive performance. **b)** Analyzing the selective prediction metrics trends, we observe larger variance for each baseline. In particular, **MedCertAIn** shows the most stable signal across seeds, showing an enhanced capability of detecting high-uncertainty patients. **c)** The difference between **MedCertAIn** and the baseline MedFuse is even more significant in selective prediction metrics.

## 5 Results

### 5.1 MedCertAIn Achieves State-Of-The-Art Performance and Reliable Selective Prediction

Table 2 reports 48h in-hospital mortality prediction results for the deterministic baselines DrFuse (Yao et al., 2024) and MedFuse (Hayat et al., 2022), a Bayesian neural network with a naive Gaussian prior (Blundell et al., 2015; Rudner et al., 2022), and our proposed method **MedCertAIn**. Overall, **MedCertAIn** achieves the strongest standard predictive performance, with AUROC $0.835 \pm 0.001$ and AUPRC $0.498 \pm 0.002$. Relative to the uninformative Gaussian prior BNN, this corresponds to a 1.83% gain in AUROC and a 2.68% gain in AUPRC, indicating improved discrimination and stronger performance under class imbalance, which is critical for reliable mortality prediction.

Beyond standard metrics, selective prediction is essential in clinical workflows, enabling models to flag uncertain cases for clinician review and improving safety under deployment. **MedCertAIn** consistently improves selective performance, achieving selective AUROC $0.857 \pm 0.005$ and selective AUPRC $0.599 \pm 0.002$. Compared to the uninformative Gaussian prior BNN, this yields a 20.5% improvement in selective AUROC and a 67.8% increase in selective AUPRC. Plots in Figure 3 show that our method is the most stable and performing across baselines for selective prediction. **MedCertAIn** provides both higher predictive performance and more reliable uncertainty estimates, facilitating clinician–AI cooperation by deferring cases that are more likely to be misclassified and supporting more efficient allocation of clinical resources.

Table 2: `MedCertAIn` **Achieves State-Of-The-Art Performance and Reliable Selective Prediction**: Quantitative performance in 48h in-hospital mortality task of across deterministic and naive stochastic baselines compared to our proposed method `MedCertAIn` which consistently beats all other models with significant improvement in selective prediction metrics.

| Metric / Model | AUROC | AUPRC | Selective AUROC | Selective AUPRC |
|---|---|---|---|---|
| MedFuse (Hayat et al., 2022) | $0.819_{\pm 0.000}$ | $0.466_{\pm 0.002}$ | $0.659_{\pm 0.005}$ | $0.203_{\pm 0.006}$ |
| Temp. Scaling (Guo et al., 2017) | $0.820_{\pm 0.000}$ | $0.470_{\pm 0.002}$ | $0.657_{\pm 0.006}$ | $0.206_{\pm 0.011}$ |
| Deep Ensemble (Lakshminarayanan et al., 2016) | $0.819_{\pm 0.000}$ | $0.469_{\pm 0.000}$ | $0.657_{\pm 0.000}$ | $0.208_{\pm 0.000}$ |
| MC Dropout (Gal & Ghahramani, 2016) | $\underline{0.822}_{\pm 0.002}$ | $0.480_{\pm 0.008}$ | $0.681_{\pm 0.007}$ | $0.233_{\pm 0.014}$ |
| Uninformative Prior (Rudner et al., 2022) | $0.820_{\pm 0.004}$ | $\underline{0.485}_{\pm 0.006}$ | $0.711_{\pm 0.034}$ | $0.357_{\pm 0.058}$ |
| `MedCertAIn` (**Ours**) | $\mathbf{0.835}_{\pm 0.001}$ | $\mathbf{0.498}_{\pm 0.002}$ | $\mathbf{0.857}_{\pm 0.005}$ | $\mathbf{0.599}_{\pm 0.002}$ |
| Improvement vs MedFuse (%) | +2.0% | +6.9% | +30.0% | +195.1% |
| Improvement vs Uninformative Prior (%) | +1.8% | +2.7% | +20.5% | +67.8% |

Table 3: **Comparison Across Informative Multimodal Data-Driven Priors.** Quantitative performance across metrics for all different model ablations of data samples with high uncertainty.

| Ablation | AUROC | AUPRC | Selective AUROC | Selective AUPRC |
|---|---|---|---|---|
| Uninformative Prior | $0.820_{\pm 0.004}$ | $0.485_{\pm 0.006}$ | $0.711_{\pm 0.034}$ | $0.357_{\pm 0.058}$ |
| Random Corruptions | $0.832_{\pm 0.001}$ | $0.494_{\pm 0.002}$ | $0.841_{\pm 0.006}$ | $0.579_{\pm 0.011}$ |
| Inter Similarity | $0.821_{\pm 0.001}$ | $0.470_{\pm 0.002}$ | $0.749_{\pm 0.013}$ | $0.423_{\pm 0.032}$ |
| Inter-Intra Similarity | $0.821_{\pm 0.001}$ | $0.470_{\pm 0.002}$ | $0.755_{\pm 0.007}$ | $0.434_{\pm 0.020}$ |
| `MedCertAIn I` | $\mathbf{0.835}_{\pm 0.001}$ | $\mathbf{0.498}_{\pm 0.002}$ | $\mathbf{0.857}_{\pm 0.005}$ | $\mathbf{0.599}_{\pm 0.002}$ |
| `MedCertAIn II` | $0.834_{\pm 0.001}$ | $0.497_{\pm 0.001}$ | $0.848_{\pm 0.008}$ | $0.590_{\pm 0.006}$ |

## 5.2 Comparison Across Informative Multimodal Data-Driven Priors

We perform ablations over high-uncertainty sample selection strategies to quantify the impact of our data-driven priors. Table 3 compares a naive Gaussian prior, random corruptions, inter-modal similarity, inter-intra modal similarity, and our proposed priors `MedCertAIn I` (inter-modal + corruptions) and `MedCertAIn II` (inter-intra + corruptions). `MedCertAIn I` corresponds to the main `MedCertAIn` variant reported in this study, as it achieves the strongest overall performance.

Gaussian priors achieve AUROC 0.820±0.004 and selective AUPRC 0.357±0.058, while random corruptions improve both standard and selective performance (AUROC 0.832±0.001, selective AUPRC 0.579±0.011). Similarity-based selection alone yields comparable standard performance (AUROC 0.821±0.001) but lower selective metrics. In contrast, `MedCertAIn I` and `MedCertAIn II` outperform all alternatives across metrics, with `MedCertAIn I` achieving the best results (AUROC 0.835±0.001, AUPRC 0.498±0.002, selective AUROC 0.857±0.005, selective AUPRC 0.599±0.002). A comparison of selective prediction trends across ablation is shown in Figure B.1. Overall, combining latent-space divergence with corruptions produces the most informative priors with strongest reliability for clinical prediction.

## 5.3 Impact of Multimodal Fusion on Selective Prediction

To quantify the benefit of fusion, Table 4 compares unimodal (CXR-only, EHR-only) and multimodal performance for the deterministic MedFuse baseline and our stochastic `MedCertAIn` framework. For `MedCertAIn`, unimodal variants use single-modality Random Corruptionncorruption context sets, excluding latent-space sampling for a fair comparison with MedFuse. Across both frameworks, EHR provides the strongest unimodal signal (MedFuse: AUROC 0.760, selective AUROC 0.670; `MedCertAIn`: AUROC 0.825, selective AUROC

Table 4: **Impact of Multimodal Fusion on Selective Prediction**: Quantitative performance across uni-modal and multimodal settings, with consistent best performance across settings obtained with `MedCertAIn`.

| Modality | Model | AUROC | AUPRC | Selective AUROC | Selective AUPRC |
|---|---|---|---|---|---|
| Unimodal CXR | MedFuse | $0.615_{\pm 0.007}$ | $0.225_{\pm 0.007}$ | $\mathbf{0.569_{\pm 0.012}}$ | $\mathbf{0.167_{\pm 0.015}}$ |
| | `MedCertAIn` | $\mathbf{0.632_{\pm 0.014}}$ | $\mathbf{0.242_{\pm 0.014}}$ | $0.567_{\pm 0.015}$ | $0.146_{\pm 0.004}$ |
| Unimodal EHR | MedFuse | $0.760_{\pm 0.003}$ | $0.393_{\pm 0.008}$ | $0.670_{\pm 0.017}$ | $0.235_{\pm 0.007}$ |
| | `MedCertAIn` | $\mathbf{0.825_{\pm 0.004}}$ | $\mathbf{0.480_{\pm 0.004}}$ | $\mathbf{0.807_{\pm 0.014}}$ | $\mathbf{0.499_{\pm 0.023}}$ |
| Multimodal | MedFuse | $0.819_{\pm 0.000}$ | $0.466_{\pm 0.002}$ | $0.659_{\pm 0.005}$ | $0.203_{\pm 0.006}$ |
| | `MedCertAIn` | $\mathbf{0.832_{\pm 0.001}}$ | $\mathbf{0.494_{\pm 0.002}}$ | $\mathbf{0.841_{\pm 0.006}}$ | $\mathbf{0.579_{\pm 0.011}}$ |

Table 5: **Analysis of `MedCertAIn` Across Patient Subpopulations.** `MedCertAIn` consistently performs better for each individual subgroup, particularly showing significant improvements in selective AUROC and selective AUPRC, demonstrating it's potential for deployment in varied clinical scenarios.

| Patient Subgroup | MedFuse | | | | MedCertAIn | | | |
|---|---|---|---|---|---|---|---|---|
| | AUROC | AUPRC | Selective AUROC | Selective AUPRC | AUROC | AUPRC | Selective AUROC | Selective AUPRC |
| **Age** | | | | | | | | |
| 18-45 years | $0.803_{\pm 0.001}$ | $\mathbf{0.603_{\pm 0.023}}$ | $0.559_{\pm 0.002}$ | $0.262_{\pm 0.001}$ | $\mathbf{0.824_{\pm 0.001}}$ | $0.600_{\pm 0.002}$ | $\mathbf{0.723_{\pm 0.005}}$ | $\mathbf{0.387_{\pm 0.017}}$ |
| 45-60 years | $0.788_{\pm 0.001}$ | $\mathbf{0.545_{\pm 0.027}}$ | $0.554_{\pm 0.003}$ | $0.231_{\pm 0.001}$ | $\mathbf{0.811_{\pm 0.002}}$ | $0.485_{\pm 0.002}$ | $\mathbf{0.730_{\pm 0.008}}$ | $\mathbf{0.299_{\pm 0.014}}$ |
| >60 years | $0.814_{\pm 0.000}$ | $\mathbf{0.528_{\pm 0.032}}$ | $0.631_{\pm 0.004}$ | $0.207_{\pm 0.002}$ | $\mathbf{0.832_{\pm 0.000}}$ | $0.480_{\pm 0.002}$ | $\mathbf{0.863_{\pm 0.004}}$ | $\mathbf{0.559_{\pm 0.002}}$ |
| **Sex** | | | | | | | | |
| Male | $0.792_{\pm 0.000}$ | $0.401_{\pm 0.001}$ | $0.603_{\pm 0.003}$ | $0.186_{\pm 0.001}$ | $\mathbf{0.810_{\pm 0.001}}$ | $\mathbf{0.425_{\pm 0.002}}$ | $\mathbf{0.832_{\pm 0.005}}$ | $\mathbf{0.494_{\pm 0.005}}$ |
| Female | $0.806_{\pm 0.000}$ | $0.420_{\pm 0.001}$ | $0.609_{\pm 0.003}$ | $0.183_{\pm 0.000}$ | $\mathbf{0.827_{\pm 0.001}}$ | $\mathbf{0.453_{\pm 0.002}}$ | $\mathbf{0.842_{\pm 0.005}}$ | $\mathbf{0.484_{\pm 0.005}}$ |

0.807), while CXR-only performance remains lower (MedFuse: AUROC 0.615; `MedCertAIn`: AUROC 0.632). Combining both modalities yields consistent gains: MedFuse improves to AUROC 0.819 and AUPRC 0.466, while `MedCertAIn` further increases to AUROC 0.832 and AUPRC 0.494. The largest improvements appear in selective prediction, where multimodal `MedCertAIn` achieves selective AUROC 0.841 and selective AUPRC 0.579, outperforming both EHR-only (0.807/0.499) and CXR-only (0.567/0.146) variants.

### 5.4 Analysis of `MedCertAIn` Across Patient Subpopulations

To evaluate robustness, we analyse performance across patient subgroups defined by age and sex. Table 5 shows that `MedCertAIn` consistently outperforms the deterministic MedFuse baseline across all subgroups, with the largest gains in selective metrics. For age groups, `MedCertAIn` improves AUROC from 0.803 to 0.824 (18–45), 0.788 to 0.811 (45–60), and 0.814 to 0.832 (>60), while selective AUROC increases from 0.559 to 0.723, 0.554 to 0.730, and 0.631 to 0.863, respectively. Although standard AUPRC is slightly lower in some age groups, selective AUPRC is consistently higher, improving from 0.262 to 0.387 (18–45), 0.231 to 0.299 (45–60), and 0.207 to 0.559 (>60).

Across sex subgroups, `MedCertAIn` also improves performance for both male and female patients. For males, AUROC increases from 0.792 to 0.810 and selective AUROC from 0.603 to 0.832, with selective AUPRC improving from 0.186 to 0.494. For females, AUROC increases from 0.806 to 0.827 and selective AUROC from 0.609 to 0.842, with selective AUPRC improving from 0.183 to 0.484. A visual comparison is provided in Figure 4. Overall, these results indicate that `MedCertAIn` maintains strong standard performance while providing substantially more reliable selective behavior, supporting its use in high-risk populations and clinically realistic deferral settings. In Appendix A, we provide complementary results across additional mortality horizons, including 3-month, 6-month, and 1-year mortality prediction, to further assess whether the observed gains in selective prediction persist beyond the 48-hour setting.

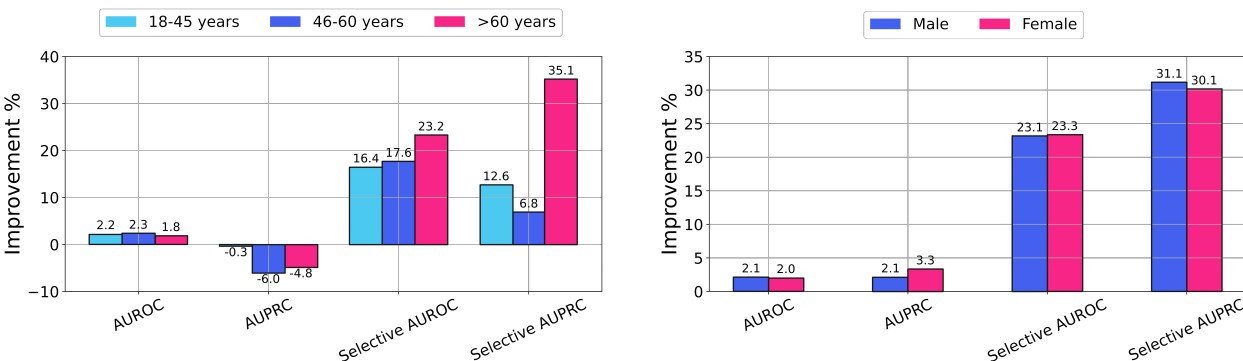

Figure 4: **Analysis of `MedCertAIn` Across Patient Subpopulations.** Comparison of percentage improvement from `MedCertAIn` over the deterministic baseline, MedFuse. We observe that our stochastic framework significantly improves selective prediction showing its adaptability over different patient subpopulation groups.

## 6 Discussion

Recent studies show that UQ methods in healthcare largely rely on Bayesian approaches but are often limited to unimodal imaging settings (Seoni et al., 2023; Lechuga et al., 2025). Our work addresses this gap by applying Bayesian UQ to multimodal clinical data combining medical images and time-series signals. A key strength of `MedCertAIn` is that context point selection is performed in an unsupervised manner (i.e., without labels), making it scalable across tasks and modalities while reducing design bias. Across experiments and subpopulation analyses, `MedCertAIn` consistently improves predictive performance and selective prediction metrics over deterministic multimodal baselines, yielding more reliable uncertainty estimates.

Despite these promising results, several limitations remain. First, our experiments focus on in-hospital mortality prediction, and generalization to other clinical tasks requires further study. Second, while we integrate imaging and time-series data, incorporating additional modalities could improve applicability and performance (Salvi et al., 2023). Although the empirical gains from cross-modal similarity are modest in the current setting, this signal remains useful because it provides a label-free, reusable, and clinically meaningful proxy for multimodal disagreement. Stronger task-adapted or clinically fine-tuned multimodal self-supervised models may further improve the utility of similarity-based context selection. Finally, reliance on the MIMIC dataset limits validity across institutions and patient populations, motivating evaluation in more diverse clinical settings (Meng et al., 2022).

Beyond performance gains, our findings suggest that uncertainty-aware multimodal models may offer practical benefits for clinical decision-making. Improved selective prediction performance indicates the potential to defer uncertain cases, which is particularly important in high-risk ICU settings. Consistent improvements across subpopulations further suggest that Bayesian approaches may contribute to more reliable model behavior across patient groups, though this warrants deeper fairness-focused investigation. Building on these results, future work should explore additional modalities such as radiology reports, extend evaluation to other clinical tasks (e.g., decompensation or length of stay), and investigate alternative fusion strategies including late fusion and missing-modality settings. Strengthening the multimodal backbone architecture and exploring self-supervised methods for high-uncertainty samples may further improve uncertainty estimation.

## 7 Conclusion

We introduced `MedCertAIn`, a Bayesian uncertainty quantification framework for multimodal in-hospital mortality prediction that combines chest X-ray imaging and EHR time-series data. Across experiments, our approach consistently improves predictive performance and uncertainty quality compared to deterministic multimodal baselines, enabling more reliable selective prediction. These results demonstrate the potential of Bayesian multimodal learning to provide better uncertainty estimates in complex clinical settings. Overall, `MedCertAIn` represents a step toward more trustworthy multimodal AI systems capable of supporting safe clinical decision-making in ICU environments.

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

# Appendix

## Table of Contents

# A Extended Results

## A.1 48h In-Hospital Mortality

Table A.1: **48h Mortality Calibration.** Calibration performance across MedFuse-based baselines and `MedCertAIn` for 48-hour mortality prediction. `MedCertAIn` achieves the lowest Brier score, while the deterministic MedFuse baseline obtains the lowest ECE.

| Metric
Model | ECE | Brier |
|---|---|---|
| **MedFuse Variations** | | |
| MedFuse (Hayat et al., 2022) | $\mathbf{3.215}_{\pm 0.242}$ | $\underline{0.102}_{\pm 0.000}$ |
| Temp. Scaling (Guo et al., 2017) | $3.527_{\pm 1.225}$ | $0.102_{\pm 0.001}$ |
| Deep Ensemble (Lakshminarayanan et al., 2016) | $\underline{3.455}_{\pm 0.000}$ | $0.102_{\pm 0.000}$ |
| MC Dropout (Gal & Ghahramani, 2016) | $3.613_{\pm 0.424}$ | $0.102_{\pm 0.001}$ |
| Uninformative Prior (Rudner et al., 2022) | $4.625_{\pm 0.929}$ | $0.102_{\pm 0.001}$ |
| `MedCertAIn` (**Ours**) | $4.323_{\pm 0.146}$ | $\mathbf{0.100}_{\pm 0.000}$ |

For 48-hour mortality prediction, `MedCertAIn` obtains the best Brier score, suggesting improved overall probabilistic accuracy compared with the MedFuse-based baselines. However, MedFuse achieves the lowest ECE and `MedCertAIn` the second largest, indicating that `MedCertAIn`'s gains in predictive reliability are not uniformly reflected across all calibration metrics. Overall, these results suggest that `MedCertAIn` improves calibration under the Brier score while maintaining competitive ECE performance.

## A.2 3-month In-Hospital Mortality

Table A.2: **3-Month Mortality Prediction Performance.** Performance comparison across MedFuse-based baselines and MedCertAIn for 3-month mortality prediction. MedCertAIn achieves the best performance across all standard and selective prediction metrics, with the largest gains observed in selective AUPRC.

| Metric / Model | AUROC | AUPRC | Selective AUROC | Selective AUPRC |
|---|---|---|---|---|
| **MedFuse Variations** | | | | |
| MedFuse (Hayat et al., 2022) | $0.656_{\pm 0.004}$ | $\underline{0.174_{\pm 0.006}}$ | $0.582_{\pm 0.009}$ | $0.078_{\pm 0.002}$ |
| Temp. Scaling (Guo et al., 2017) | $0.656_{\pm 0.004}$ | $\underline{0.174_{\pm 0.006}}$ | $0.582_{\pm 0.009}$ | $0.078_{\pm 0.002}$ |
| Deep Ensemble (Lakshminarayanan et al., 2016) | $\underline{0.657_{\pm 0.000}}$ | $0.180_{\pm 0.000}$ | $\underline{0.583_{\pm 0.000}}$ | $0.077_{\pm 0.000}$ |
| MC Dropout (Gal & Ghahramani, 2016) | $0.656_{\pm 0.004}$ | $0.175_{\pm 0.010}$ | $0.556_{\pm 0.013}$ | $\underline{0.102_{\pm 0.002}}$ |
| Uninformative Prior (Rudner et al., 2022) | $0.651_{\pm 0.021}$ | $0.146_{\pm 0.021}$ | $0.581_{\pm 0.031}$ | $0.079_{\pm 0.003}$ |
| **MedCertAIn (Ours)** | $\mathbf{0.660_{\pm 0.001}}$ | $\mathbf{0.183_{\pm 0.009}}$ | $\mathbf{0.683_{\pm 0.020}}$ | $\mathbf{0.196_{\pm 0.018}}$ |
| Improvement vs MedFuse (%) | +0.6% | +5.2% | +17.4% | +151.3% |
| Improvement vs Uninformative Prior (%) | +1.4% | +25.3% | +17.6% | +148.1% |

Table A.3: **3-Month Mortality Calibration.** Calibration performance across MedFuse-based baselines and MedCertAIn for 3-month mortality prediction. Deep ensembles and MC Dropout obtain the strongest calibration scores, while MedCertAIn shows weaker calibration despite outperforming all baselines in predictive and selective prediction metrics.

| Metric / Model | ECE | Brier |
|---|---|---|
| **MedFuse Variations** | | |
| MedFuse (Hayat et al., 2022) | $0.979_{\pm 0.347}$ | $\mathbf{0.068_{\pm 0.000}}$ |
| Temp. Scaling (Guo et al., 2017) | $1.545_{\pm 0.137}$ | $\underline{0.069_{\pm 0.000}}$ |
| Deep Ensemble (Lakshminarayanan et al., 2016) | $\mathbf{0.847_{\pm 0.000}}$ | $\mathbf{0.068_{\pm 0.000}}$ |
| MC Dropout (Gal & Ghahramani, 2016) | $\underline{0.937_{\pm 0.347}}$ | $\mathbf{0.068_{\pm 0.000}}$ |
| Uninformative Prior (Rudner et al., 2022) | $1.313_{\pm 1.036}$ | $0.070_{\pm 0.004}$ |
| **MedCertAIn (Ours)** | $1.492_{\pm 0.049}$ | $0.076_{\pm 0.001}$ |

For 3-month mortality prediction, MedCertAIn outperforms all baselines across AUROC, AUPRC, selective AUROC, and selective AUPRC. The improvement is especially large for selective AUPRC, indicating that MedCertAIn better identifies uncertain cases that should be deferred for review. In contrast, calibration metrics show a different trend: Deep ensembles and MC Dropout achieve the lowest ECE and Brier scores, while MedCertAIn is less competitive on calibration. These results highlight a trade-off in which MedCertAIn substantially improves selective prediction reliability, even though it does not dominate calibration for this horizon.

### A.3 6-month In-Hospital Mortality

Table A.4: **6-Month Mortality Prediction Performance.** Performance comparison across MedFuse-based baselines and `MedCertAIn` for 6-month mortality prediction. `MedCertAIn` achieves the best AUROC and substantially improves selective prediction performance, with the largest gains observed in selective AUROC and selective AUPRC.

| Metric
Model | AUROC | AUPRC | Selective
AUROC | Selective
AUPRC |
|---|---|---|---|---|
| **MedFuse Variations** | | | | |
| MedFuse (Hayat et al., 2022) | $0.642_{\pm0.005}$ | $0.196_{\pm0.013}$ | $0.544_{\pm0.011}$ | $0.102_{\pm0.002}$ |
| Temp. Scaling (Guo et al., 2017) | $0.642_{\pm0.005}$ | $0.196_{\pm0.013}$ | $0.544_{\pm0.011}$ | $0.102_{\pm0.002}$ |
| Deep Ensemble (Lakshminarayanan et al., 2016) | $0.646_{\pm0.000}$ | $\mathbf{0.211_{\pm0.000}}$ | $0.544_{\pm0.000}$ | $0.100_{\pm0.000}$ |
| MC Dropout (Gal & Ghahramani, 2016) | $0.648_{\pm0.002}$ | $\underline{0.205_{\pm0.006}}$ | $0.556_{\pm0.013}$ | $0.102_{\pm0.002}$ |
| Uninformative Prior (Rudner et al., 2022) | $0.644_{\pm0.003}$ | $0.197_{\pm0.009}$ | $\underline{0.563_{\pm0.013}}$ | $\underline{0.103_{\pm0.003}}$ |
| `MedCertAIn`(**Ours**) | $\mathbf{0.654_{\pm0.005}}$ | $0.189_{\pm0.006}$ | $\mathbf{0.683_{\pm0.020}}$ | $\mathbf{0.196_{\pm0.027}}$ |
| Improvement vs MedFuse (%) | +1.9% | -3.6% | +25.6% | +92.2% |
| Improvement vs Uninformative Prior (%) | +1.6% | -4.1% | +21.3% | +90.3% |

Table A.5: **6-Month Mortality Calibration.** Calibration performance across MedFuse-based baselines and `MedCertAIn` for 6-month mortality prediction. Deep ensembles obtain the lowest ECE, while several baselines achieve comparable Brier scores; `MedCertAIn` remains competitive but does not improve calibration for this horizon.

| Metric
Model | ECE | Brier |
|---|---|---|
| **MedFuse Variations** | | |
| MedFuse (Hayat et al., 2022) | $1.168_{\pm0.451}$ | $\underline{0.092_{\pm0.000}}$ |
| Temp. Scaling (Guo et al., 2017) | $\underline{0.831_{\pm0.808}}$ | $0.092_{\pm0.000}$ |
| Deep Ensemble (Lakshminarayanan et al., 2016) | $\mathbf{0.699_{\pm0.000}}$ | $\mathbf{0.091_{\pm0.000}}$ |
| MC Dropout (Gal & Ghahramani, 2016) | $1.452_{\pm0.419}$ | $\mathbf{0.091_{\pm0.000}}$ |
| Uninformative Prior (Rudner et al., 2022) | $2.065_{\pm0.149}$ | $\mathbf{0.091_{\pm0.000}}$ |
| `MedCertAIn`(**Ours**) | $2.105_{\pm0.108}$ | $\underline{0.092_{\pm0.000}}$ |

For 6-month mortality prediction, `MedCertAIn` achieves the highest AUROC and substantially improves selective prediction performance compared with MedFuse and the uninformative prior. The strongest gains are observed in selective AUROC and selective AUPRC, indicating improved ability to identify uncertain cases for deferral. However, `MedCertAIn` does not improve AUPRC and is less competitive on calibration metrics, where deep ensembles and other baselines obtain lower ECE or Brier scores. These results suggest that `MedCertAIn` mainly improves uncertainty-aware selective prediction at this horizon, while standard precision-recall performance and calibration remain more mixed.

**A.4   1-year In-Hospital Mortality**

Table A.6: **1-Year Mortality Prediction Performance.** Performance comparison across MedFuse-based baselines and `MedCertAIn` for 1-year mortality prediction. `MedCertAIn` achieves the best AUROC and selective prediction performance, with particularly large improvements in selective AUPRC.

| Metric / Model | AUROC | AUPRC | Selective AUROC | Selective AUPRC |
|---|---|---|---|---|
| **MedFuse Variations** | | | | |
| MedFuse (Hayat et al., 2022) | $0.636_{\pm 0.005}$ | $0.227_{\pm 0.007}$ | $0.545_{\pm 0.009}$ | $0.137_{\pm 0.002}$ |
| Temp. Scaling (Guo et al., 2017) | $0.636_{\pm 0.005}$ | $0.227_{\pm 0.007}$ | $0.545_{\pm 0.009}$ | $0.137_{\pm 0.002}$ |
| Deep Ensemble (Lakshminarayanan et al., 2016) | $\underline{0.638_{\pm 0.000}}$ | $\underline{0.230_{\pm 0.000}}$ | $0.544_{\pm 0.000}$ | $0.136_{\pm 0.000}$ |
| MC Dropout (Gal & Ghahramani, 2016) | $0.637_{\pm 0.003}$ | $\mathbf{0.231_{\pm 0.005}}$ | $0.545_{\pm 0.014}$ | $0.135_{\pm 0.001}$ |
| Uninformative Prior (Rudner et al., 2022) | $0.593_{\pm 0.023}$ | $0.197_{\pm 0.014}$ | $0.537_{\pm 0.031}$ | $0.167_{\pm 0.020}$ |
| `MedCertAIn` (**Ours**) | $\mathbf{0.639_{\pm 0.004}}$ | $0.222_{\pm 0.005}$ | $\mathbf{0.661_{\pm 0.011}}$ | $\mathbf{0.252_{\pm 0.015}}$ |
| Improvement vs MedFuse (%) | +0.5% | -2.2% | +21.3% | +83.9% |
| Improvement vs Uninformative Prior (%) | +7.8% | +12.7% | +23.1% | +50.9% |

Table A.7: **1-Year Mortality Calibration.** Calibration performance across MedFuse-based baselines and `MedCertAIn` for 1-year mortality prediction. MC Dropout achieves the lowest ECE, while deterministic and ensemble baselines obtain the lowest Brier scores; `MedCertAIn` remains close in Brier score but does not dominate calibration.

| Metric / Model | ECE | Brier |
|---|---|---|
| **MedFuse Variations** | | |
| MedFuse (Hayat et al., 2022) | $2.086_{\pm 0.430}$ | $\mathbf{0.118_{\pm 0.000}}$ |
| Temp. Scaling (Guo et al., 2017) | $\underline{2.007_{\pm 0.502}}$ | $\mathbf{0.118_{\pm 0.000}}$ |
| Deep Ensemble (Lakshminarayanan et al., 2016) | $2.718_{\pm 0.000}$ | $\mathbf{0.118_{\pm 0.000}}$ |
| MC Dropout (Gal & Ghahramani, 2016) | $\mathbf{1.879_{\pm 0.622}}$ | $\mathbf{0.118_{\pm 0.000}}$ |
| Uninformative Prior (Rudner et al., 2022) | $5.447_{\pm 2.330}$ | $0.135_{\pm 0.011}$ |
| `MedCertAIn` (**Ours**) | $2.399_{\pm 0.120}$ | $\underline{0.119_{\pm 0.001}}$ |

For 1-year mortality prediction, `MedCertAIn` achieves the best AUROC and substantially improves both selective AUROC and selective AUPRC. Although its AUPRC is slightly lower than several MedFuse-based baselines, `MedCertAIn` shows strong gains in selective prediction, suggesting better ranking of uncertain cases for clinical review. Calibration results again show a different trend: MC Dropout obtains the lowest ECE, and several baselines achieve slightly better Brier scores. Overall, these results reinforce that `MedCertAIn` provides the largest benefits in selective prediction reliability, particularly for longer-horizon mortality risk.

## A.5 Comparison Across Deterministic Multimodal Models

Table A.8: **Comparison of Deterministic Multimodal Models Across Mortality Prediction Horizons.** Quantitative performance of DrFuse, MeTra, and MedFuse across 48-hour, 3-month, 6-month, and 1-year mortality prediction tasks. We report standard predictive performance using AUROC and AUPRC, selective prediction performance using selective AUROC and selective AUPRC, and calibration using expected calibration error (ECE) and Brier score. Results are reported as mean ± standard deviation across 5 seed runs. The best-performing model for each task and metric is highlighted in bold, and the second-best result is underlined.

| Model | AUROC | AUPRC | Selective AUROC | Selective AUPRC | ECE ↓ | Brier ↓ |
|---|---|---|---|---|---|---|
| **48h Mortality** | | | | | | |
| DrFuse (Yao et al., 2024) | $0.817_{\pm 0.005}$ | $\mathbf{0.472}_{\pm 0.013}$ | $\mathbf{0.785}_{\pm 0.007}$ | $\mathbf{0.437}_{\pm 0.035}$ | $8.943_{\pm 0.597}$ | $0.111_{\pm 0.006}$ |
| MeTra (Khader et al., 2023) | $0.783_{\pm 0.015}$ | $0.411_{\pm 0.037}$ | $0.724_{\pm 0.034}$ | $0.296_{\pm 0.083}$ | $\mathbf{2.062}_{\pm 0.918}$ | $0.110_{\pm 0.005}$ |
| MedFuse (Hayat et al., 2022) | $\mathbf{0.819}_{\pm 0.000}$ | $0.466_{\pm 0.002}$ | $0.659_{\pm 0.005}$ | $0.203_{\pm 0.006}$ | $3.215_{\pm 0.242}$ | $\mathbf{0.102}_{\pm 0.000}$ |
| **3-month Mortality** | | | | | | |
| DrFuse (Yao et al., 2024) | $0.647_{\pm 0.039}$ | $0.150_{\pm 0.035}$ | $0.574_{\pm 0.034}$ | $0.091_{\pm 0.024}$ | $4.159_{\pm 2.585}$ | $0.079_{\pm 0.018}$ |
| MeTra (Khader et al., 2023) | $0.596_{\pm 0.028}$ | $0.109_{\pm 0.015}$ | $\mathbf{0.588}_{\pm 0.038}$ | $0.085_{\pm 0.008}$ | $1.756_{\pm 0.924}$ | $0.071_{\pm 0.002}$ |
| MedFuse (Hayat et al., 2022) | $\mathbf{0.656}_{\pm 0.004}$ | $\mathbf{0.174}_{\pm 0.006}$ | $0.582_{\pm 0.009}$ | $0.078_{\pm 0.002}$ | $\mathbf{0.979}_{\pm 0.347}$ | $\mathbf{0.068}_{\pm 0.000}$ |
| **6-month Mortality** | | | | | | |
| DrFuse (Yao et al., 2024) | $\mathbf{0.687}_{\pm 0.011}$ | $\mathbf{0.230}_{\pm 0.008}$ | $0.551_{\pm 0.018}$ | $0.098_{\pm 0.003}$ | $2.677_{\pm 1.512}$ | $\mathbf{0.090}_{\pm 0.002}$ |
| MeTra (Khader et al., 2023) | $0.594_{\pm 0.030}$ | $0.146_{\pm 0.015}$ | $\mathbf{0.562}_{\pm 0.044}$ | $\mathbf{0.112}_{\pm 0.007}$ | $2.966_{\pm 1.530}$ | $0.095_{\pm 0.003}$ |
| MedFuse (Hayat et al., 2022) | $0.642_{\pm 0.005}$ | $0.196_{\pm 0.013}$ | $0.544_{\pm 0.011}$ | $0.102_{\pm 0.002}$ | $\mathbf{1.168}_{\pm 0.451}$ | $0.092_{\pm 0.000}$ |
| **1-year Mortality** | | | | | | |
| DrFuse (Yao et al., 2024) | $\mathbf{0.663}_{\pm 0.009}$ | $\mathbf{0.257}_{\pm 0.005}$ | $0.572_{\pm 0.015}$ | $0.134_{\pm 0.003}$ | $3.501_{\pm 2.808}$ | $0.119_{\pm 0.005}$ |
| MeTra (Khader et al., 2023) | $0.626_{\pm 0.002}$ | $0.202_{\pm 0.010}$ | $\mathbf{0.595}_{\pm 0.021}$ | $\mathbf{0.144}_{\pm 0.005}$ | $\mathbf{1.906}_{\pm 1.514}$ | $0.119_{\pm 0.001}$ |
| MedFuse (Hayat et al., 2022) | $0.636_{\pm 0.005}$ | $0.227_{\pm 0.007}$ | $0.545_{\pm 0.009}$ | $0.137_{\pm 0.002}$ | $2.086_{\pm 0.430}$ | $\mathbf{0.118}_{\pm 0.000}$ |

Across deterministic baselines, no single model dominates all mortality horizons and evaluation criteria. MedFuse generally achieves the strongest standard predictive performance for short- and medium-term mortality, particularly for 48-hour and 3-month mortality, while DrFuse performs best on several longer-horizon for standard AUROC/AUPRC metrics. However, selective prediction performance varies substantially across models: DrFuse performs best for selective metrics at 48 hours, whereas MeTra often achieves stronger selective AUROC or selective AUPRC at longer horizons. Calibration results also show a different trend, with MedFuse often obtaining the lowest Brier score and MeTra or MedFuse achieving the lowest ECE depending on the horizon. Overall, these results suggest that deterministic multimodal models exhibit trade-offs between discrimination, selective reliability, and calibration, motivating the need for uncertainty-aware approaches such as MedCertAIn.

## A.6 Comparison with Hard Example Mining

Table A.9: **HEM Performance**: Performance comparison of deterministic and stochastic models using Hard Example Mining for fine-tuning during training.

| Hard Example Mining (HEM) | | | | |
|---|---|---|---|---|
| **Model** | **AUROC** | **AUPRC** | **Selective AUROC** | **Selective AUPRC** |
| MedFuse (HEM (Shrivastava et al., 2016a)) | $0.233_{\pm0.003}$ | $0.094_{\pm0.001}$ | $0.393_{\pm0.010}$ | $0.070_{\pm0.010}$ |
| MedCertAIn (HEM 20%) | $\mathbf{0.821}_{\pm\mathbf{0.002}}$ | $\mathbf{0.477}_{\pm\mathbf{0.002}}$ | $\mathbf{0.790}_{\pm\mathbf{0.011}}$ | $\mathbf{0.502}_{\pm\mathbf{0.023}}$ |
| Change (%) | 58.75% | 38.30% | 39.65% | 43.26% |

We further consider Hard Example Mining (HEM), including online HEM (Shrivastava et al., 2016b) which prioritizes high-loss examples during training. In contrast, `MedCertAIn` uses high-loss samples to construct an uncertainty-aware prior rather than as training targets. We show that HEM-based fine-tuning degrades performance, while using HEM samples as context sets improves results but remains inferior to `MedCertAIn`.

The primary aim of HEM is to improve performance by focusing training on difficult samples, whereas our method targets uncertainty estimation and selective prediction, and as such, is not directly comparable to our approach. Nonetheless, to further elucidate the distinctions between `MedCertAIn` and HEM, we conducted controlled comparisons using the top 20% hardest training samples (ranked by loss). Incorporating the HEM subset as a context set within `MedCertAIn`'s uncertainty-aware prior, the model demonstrated a substantial performance boost (AUROC: 0.821, AUPRC: 0.477, Selective AUROC: 0.790, Selective AUPRC: 0.502). This represents relative gains of 58.75% AUROC, 38.30% AUPRC, 39.65% Selective AUROC, and 43.26% Selective AUPRC, underscoring the robustness of our selective prediction objective.

Unlike HEM, which emphasizes learning directly from difficult samples, `MedCertAIn` leverages them to inform uncertainty modeling while explicitly avoiding overfitting. This is enabled by our training objective, which incorporates two complementary terms: one that fits the model to the full training set, ensuring broad generalization, and another that shapes the uncertainty-aware prior using difficult examples, encouraging the model reduce its confidence on ambiguous inputs.

## B   Comparison Across Informative Multimodal Data-Driven Priors

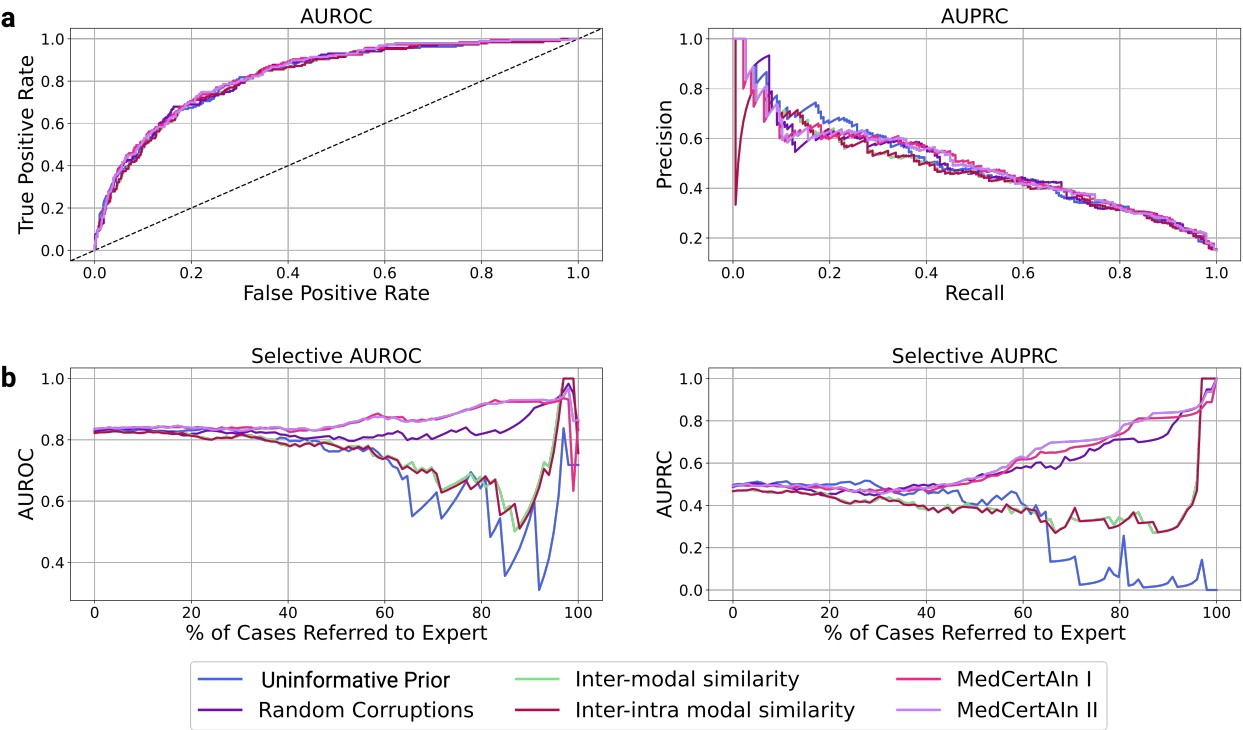

Figure B.1:   **Comparison Across Informative Multimodal Data-Driven Priors. a)** Difference across baselines in standard metrics is almost negligible. **b)** The trends in selective prediction metrics show that `MedCertAIn` remains the most performing model compared to other ablations, showing that the combination of different context-set priors enhances uncertainty-ware predictions.

# C   Derivation of Variational Objective

We adapt the approach in Rudner et al. (2024a) to the multi-modal settings. We reproduce the derivation from Rudner et al. (2024a) verbatim below.

## C.1   A Family of Data-Driven Priors

Consider a parametric observation model $p_{Y|X,\Theta}(y \,|\, x, \theta; f)$, and let the mapping $f$ be defined by $f(\cdot\,;\theta) \doteq h(\cdot\,;\theta_h)\theta_L$, where $h(\cdot\,;\theta_h)$ is the post-activation output of the penultimate layer, $\Theta_L$ is the set of stochastic final-layer parameters, $\Theta_h$ is the set of stochastic non-final-layer parameters, and $\Theta \doteq \{\Theta_h, \Theta_L\}$ is the full set of stochastic parameters. We assume access to pre-trained feature parameters, $\theta_h^*$, and context data that encodes useful information about the downstream tasks. We denote a batch of context inputs with corresponding context labels by $x_c = \{x_1, ..., x_M\}$ and $y_c = \{y_1, ..., y_M\}$, respectively, and let $p_{X_c, Y_c}$ be a joint distribution over context batches.

To construct a family of data-driven priors, we begin by specifying a *context inference problem*. We consider a Bernoulli random variable $\check{Z}$ denoting whether a given set of neural network parameters induces predictions that exhibit some desired property (e.g., high uncertainty on a set of evaluation points). Furthermore, we define a *context observation model* $\check{p}_{\check{Z}|\Theta}(\check{z} \,|\, \theta; f, p_{X_c, Y_c})$—which denotes the likelihood of observing a yet-to-be-specified outcome $\check{z}$ under $\check{p}_{\check{Z}|\Theta}$ given $\theta$ and $p_{X_c, Y_c}$—and specify a *base prior* over the model parameters, $p_\Theta(\theta)$. For notational simplicity, we will drop the subscripts going forward except when needed for clarity. With this setup, we can now define the context inference problem as finding the conditional distribution over neural network parameters that we *would* obtain if we conditioned on the desired property being satisfied. This conditional distribution will serve as our data-driven prior, and by Bayes' Theorem, we can express it as

$$p(\theta \,|\, \check{z}; p_{X_c, Y_c}) = \frac{\check{p}(\check{z} \,|\, \theta; p_{X_c, Y_c})p(\theta)}{\check{p}(\check{z}; p_{X_c, Y_c})}. \tag{C.1}$$

To define a family of data-driven priors that place high probability density on neural network parameter values that induce predictive functions with reliable uncertainty estimates, we specify a Bernoulli context observation model $\check{p}_{\check{Z}|\Theta}$ in which $\check{Z} = 1$ denotes the outcome of 'achieving reliable uncertainty quantification' and $\check{p}(\check{z} = 1 \,|\, \theta; p_{X_c, Y_c})$ denotes the likelihood of $\check{z} = 1$ given $\theta$ and $p_{X_c, Y_c}$. More specifically, we define:

$$\check{p}(\check{z} = 1 \,|\, \theta; p_{X_c, Y_c}) = \exp(-\mathbb{E}_{p_{X_c, Y_c}}[c(X_c, Y_c, \theta)])$$
$$\check{p}(\check{z} = 0 \,|\, \theta; p_{X_c, Y_c}) = 1 - \check{p}(\check{z} = 1 \,|\, \theta; p_{X_c, Y_c}), \tag{C.2}$$

where $c : \mathcal{X} \times \mathcal{Y} \times \mathbb{R}^P \to \mathbb{R}_\geq$ is a *cost function*. By specifying the outcome $\check{z} = 1$ along with a distribution, $p_{X_c, Y_c}$ we obtain a conditional distribution $\check{p}(\theta \,|\, \check{z}; f, p_{X_c, Y_c})$—the distribution over neural network parameters that we *would* infer if we observed the outcome $\check{z} = 1$ under the base prior and the Bernoulli context observation model defined in Equation (C.2). Naturally, the quality (i.e., the usefulness) of this conditional distribution is determined by the quality of the context observation model $\check{p}_{\check{Z}|\Theta}$, the data, and the prior. As a result, the primary challenge in designing effective uncertainty-aware priors lies in constructing a context observation model—via a cost function $c$ and a context distribution $p_{X_c, Y_c}$—that is as well-specified as possible. The better specified the context observation model, the more useful the data-driven prior.

## C.2   Data-Driven, Uncertainty-Aware Priors for Fine-Tuning Pre-trained Models

In this section, we present a specific instantiation of an uncertainty-aware prior for fine-tuning foundation models. To define a data-driven prior $\check{p}(\theta \,|\, \check{z}; p_{X_c, Y_c})$ that incorporates useful information from the pre-trained parameters $\theta_h^*$ and assigns high probability density to parameter values $\theta$ that induce models with reliable uncertainty quantification, we need to specify a suitable context likelihood and suitable layer-specific base priors $p(\theta_h)$ and $p(\theta_L)$. For the base priors, we let $p(\theta_h) = \mathcal{N}(\theta_h; \theta_h^*, \tau_h^{-1}I)$, which assigns high probability to parameters $\theta_h$ that are close to the pre-trained parameters $\theta_h^*$, and $p(\theta_L) = \mathcal{N}(\theta_L; \mathbf{0}, \tau_L^{-1}I)$.

To define a context observation model that induces a data-driven prior with desirable properties, we specify a cost function $c$ of the form

$$c(x_c, y_c, \theta) \doteq \tau \sum_{k=1}^{K} D_{\mathcal{M}}^2([f(x_c; \theta)]_k, m(x_c, y_c)_k, C(x_c)), \tag{C.3}$$

where $K$ is the number of output dimensions, $p_{X_c, Y_c}$ is a joint distribution over context batches $x_c$ and $y_c$ (each of size $M$),

$$D_{\mathcal{M}}^2([f(x_c; \theta)]_k, m(x_c, y_c)_k, C(x_c)) \doteq \mathbf{v}_k^\top C(x_c)^{-1} \mathbf{v}_k \tag{C.4}$$

with $\mathbf{v}_k \doteq [f(x_c; \theta)]_k - m(x_c, y_c)_k$ is the squared Mahalanobis distance between model predictions $[f(x_c; \theta)]_k$ and an input-dependent distribution with mean $m(x_c, y_c)_k$ and $M$-by-$M$ covariance matrix $C(x_c)$. To obtain a data-driven prior that assigns high probability density to parameters $\theta$ that induce models with reliable uncertainty estimates, we specify a data-dependent mean function, $m(x_c, y_c)_k \doteq [y_c]_k$, and a covariance function

$$C(\cdot) \doteq s_1 h(\cdot; \theta_h^*) h(\cdot; \theta_h^*)^\top + s_2 I, \tag{C.5}$$

parameterized by pre-trained model parameters $\theta_h^*$ and fixed scaling parameters $\tau$, $s_1$, and $s_2$, that reflects structure in the pre-trained model representations $h(\cdot; \theta_h^*)$. Finally, we define the context distribution as $p(x_c, y_c) = p(y_c \mid x_c) p(x_c)$, where

$$p(y_c \mid x_c) \doteq \delta(\{\mathbf{0}, ..., \mathbf{0}\} - y_c)$$

and $p(x_c)$ is an empirical distribution constructed from a larger set of (domain- and task-specific) context inputs.[1]

Under this cost function and context distribution, the data-driven prior defined in Equation (C.2), by design, assigns high probability density to parameters $\theta$ that induce predictions $f(x_c; \theta)$ that have high predictive uncertainty on the context inputs. If the distribution over context inputs, $p_{X_c}$, is specified to place high probability density on context batches which contain input points that are meaningfully distinct from the training inputs, then the data-driven prior favours models that exhibit high predictive uncertainty on such meaningfully distinct inputs.

## C.3 Variational Inference with Uncertainty-Aware Priors

In this section, we show how to perform variational inference with uncertainty-aware priors. We start by specifying a probabilistic model with an uncertainty-aware prior,

$$p(y_{\mathcal{D}}, \theta \mid x_{\mathcal{D}}, \check{z}; p_{X_c, Y_c}) = \underbrace{p(y_{\mathcal{D}} \mid x_{\mathcal{D}}, \theta; f)}_{\text{Likelihood}} \cdot \underbrace{p(\theta \mid \check{z}; p_{X_c, Y_c})}_{\text{Uncertainty-aware prior}} . \tag{C.6}$$

To perform variational inference in this model and approximate the posterior distribution over the parameters of interest, we begin by defining a variational distribution,

$$q(\theta) \doteq q(\theta_h) \, q(\theta_L), \tag{C.7}$$

where $q(\theta_L) = \mathcal{N}(\theta_L; \mu_L, \Sigma_L)$ and $q(\theta_h) = \mathcal{N}(\theta_h; \mu_h, \Sigma_h)$ with learnable variational parameters $\mu \doteq \{\mu_h, \mu_L\}$ and $\Sigma \doteq \{\Sigma_h, \Sigma_L\}$, and frame the inference problem of finding the posterior $p(\theta \mid x_{\mathcal{D}}, y_{\mathcal{D}}, \check{z})$ variationally as

$$\min_{q_\Theta \in \mathcal{Q}} \mathbb{D}_{\text{KL}}(q_\Theta \parallel p_{\Theta \mid X_{\mathcal{D}}, Y_{\mathcal{D}}, \check{Z}}), \tag{C.8}$$

where $\mathcal{Q}$ is a mean-field Gaussian variational family. This variational problem can equivalently be expressed as maximizing the variational objective

$$\bar{\mathcal{F}}(q_\Theta) \doteq \mathbb{E}_{q_\Theta}[\log p(y_{\mathcal{D}} \mid x_{\mathcal{D}}, \Theta; f)] - \mathbb{D}_{\text{KL}}(q_\Theta \parallel p_{\Theta \mid \check{z}}). \tag{C.9}$$

---

[1]Defining $p(y_c \mid x_c) \doteq \delta(\{\mathbf{0}, ..., \mathbf{0}\} - y_c)$ implies that under $p_{X_c, Y_c}$, all context batch samples have $y_c = \mathbf{0}$, and therefore, we effectively have $m(x_c, y_c)_k \doteq \mathbf{0}$ for all context batch samples.

Unfortunately, this variational objective is intractable since the data-driven prior $\check{p}(\theta \mid \check{z}; p_{X_c, Y_c})$ defined in Equation (C.1)—which is required to compute $\mathbb{D}_{\mathrm{KL}}(q_\Theta \parallel p_{\Theta \mid \check{z}})$—is not in general tractable.

To overcome this intractability, we take advantage of the properties of the KL divergence and note that we can express $\mathbb{D}_{\mathrm{KL}}(q_\Theta \parallel p_{\Theta \mid \check{z}})$ as:

$$\mathbb{D}_{\mathrm{KL}}(q_\Theta \parallel p_{\Theta \mid \check{z}}) = \mathbb{E}_{q_{\Theta_h} q_{\Theta_L}} \left[ \log \frac{q(\Theta_h)\, q(\Theta_L)}{p(\Theta_h)\, p(\Theta_L)} \right] - \mathbb{E}_{q_{\Theta_h} q_{\Theta_L}} [\log \check{p}(\check{z} \mid \Theta; p_{X_c, Y_c})] + \log \check{p}(\check{z}; p_{X_c, Y_c}), \quad (\text{C.10})$$

where the intractable log-marginal likelihood $\log \check{p}(\check{z}; p_{X_c, Y_c})$ was factored out as an additive constant independent of any learnable parameters. Using this result, we can obtain a tractable upper bound on the KL divergence:

$$\log \check{p}(\check{z}; p_{X_c, Y_c}) \geq \mathbb{E}_{q_{\Theta_h} q_{\Theta_L}} [\log \check{p}(\check{z} \mid \Theta; p_{X_c, Y_c})] - \mathbb{D}_{\mathrm{KL}}(q_{\Theta_h} \parallel p_{\Theta_h}) - \mathbb{D}_{\mathrm{KL}}(q_{\Theta_L} \parallel p_{\Theta_L}), \quad (\text{C.11})$$

where each KL divergence term can be computed analytically, and we can obtain an unbiased estimator of the negative log-likelihood using simple Monte Carlo estimation. Using this inequality, we can obtain a lower bound on the log marginal likelihood and obtain a valid variational objective.

**Variational Objective.** Since $q_{\Theta_h}$ and $q_{\Theta_L}$ are both mean-field Gaussian distributions, we can obtain a doubly lower bounded variational objective

$$\mathcal{F}(\mu, \Sigma) \doteq \underbrace{\mathbb{E}_{q_\Theta}[\log p(y_\mathcal{D} \mid x_\mathcal{D}, \Theta; f)]}_{\text{Expected log-likelihood}} - \underbrace{\mathbb{D}_{\mathrm{KL}}(q_{\Theta_L} \parallel p_{\Theta_L})}_{\text{Pre-training regularization}} - \underbrace{\mathbb{D}_{\mathrm{KL}}(q_{\Theta_L} \parallel p_{\Theta_L})}_{\text{Final-layer regularization}} - \underbrace{\mathbb{E}_{q_\Theta}[\mathbb{E}_{p_{X_c, Y_c}}[c(X_c, Y_c, \Theta)]]}_{\text{Uncertainty regularization}},$$
$$(\text{C.12})$$

where the cost function and context distribution are as defined above. We can estimate all expectations in the objective using simple Monte Carlo estimation, giving the final variational objective:

$$\hat{\mathcal{F}}(\mu, \Sigma) \doteq \frac{1}{J} \sum_{j=1}^{J} \log p(y_\mathcal{D} \mid x_\mathcal{D}, \theta^{(j)}; f) - \mathbb{D}_{\mathrm{KL}}(q_\Theta \parallel p_\Theta) - \frac{1}{J J'} \sum_{j=1}^{J} \sum_{j'=1}^{J'} c(x_c^{(j')}, y_c^{(j')}, \theta^{(j)}), \quad (\text{C.13})$$

with $\theta^{(j)} \sim q_\Theta$, $x_c^{(j')} \sim p_{X_c}$, and $y_c^{(j')} \sim p_{Y_c \mid X_c}$ for $j = 1, ..., J$ and $j' = 1, ..., J'$. This objective can be maximized with stochastic variational inference (Hoffman et al., 2013).

# D Experimental Details

## D.1 Dataset Training Details

Table D.10: **Summary of Original Datasets.** Number of unimodal and multimodal data points available in MIMIC-IV (EHR) and MIMIC-CXR (CXR) (Johnson et al., 2019; 2021) for the in-hospital mortality task. We note that the size of the multimodal dataset for our task decreases since we drop all data samples that do not have both modalities.

| Dataset | Training | Validation | Testing |
|---|---|---|---|
| Unimodal CXR | 124,671 | 8,813 | 20,747 |
| Unimodal EHR | 42,628 | 4,802 | 11,914 |
| Multimodal (CXR + EHR) | 4,485 | 488 | 1,242 |

## D.2 Training details of ConVIRT

Contrastive VIsual Representation Learning from Text (ConVIRT) (Zhang et al., 2022) is a vision-language self-supervised pretraining framework developed for medical images and radiology reports. ConVIRT is based on a bidirectional contrastive objective between pretraining modalities, maximizing the similarity of the latent space embeddings of an image-text pair. The loss function of our ConVIRT-MedFuse architecture is defined as $\mathcal{L} = \mathcal{S}(x^{\mathrm{ehr}}, x^{\mathrm{cxr}}) + \mathcal{S}(x^{\mathrm{cxr}}, x^{\mathrm{ehr}})$, where $\mathcal{S}$ is the infoNCE loss.

We conducted 10 hyperparameter tuning runs via random search by sampling a learning rate from a uniform distribution in the range $[10^{-1}, 10^{-2}]$. We used a batch size of 256 across experiments, setting the maximum number of epochs to 300, with no early stopping. To select the best model, we chose the best checkpoint based on the best AUROC score achieved on the validation set across epochs and models. Using the best model, we made an inference pass using the original training data $(X^{\mathrm{ehr}}, X^{\mathrm{cxr}})$ to obtain the latent space representation and compute the cosine similarity between data modalities.

## D.3 Stochastic Hyperparameter Space

Table D.11: **Hyperparameter Search Space.** Hyperparameters and corresponding value ranges used for the uncertainty regularizer of `MedCertAIn`.

| Hyperparameter | Values |
|---|---|
| prior variance | [0.1, 1, 10, 1000] |
| prior likelihood scale | [0.1, 1, 10] |
| prior likelihood f-scale | [0, 1, 10] |
| prior likelihood covariance scale | [0.1, 0.01, 0.001] |
| prior likelihood covariance diagonal | [0.5, 1, 5] |

## D.4 Technical Implementation

Our data loading and pre-processing pipeline was implemented using PyTorch, (Paszke et al., 2019) following the same structure of the code used by Hayat et al. (2022). However, we refactored the original unimodal and multimodal models, training, and evaluation loops using Jax (Bradbury et al., 2018). This framework simplifies the implementation of Bayesian neural networks and stochastic training, which are the basis of the uncertainty quantification methods used in this work. In addition, thanks to this code refactoring, we obtained a significant reduction in total training time for the unimodal and multimodal models compared to the code baseline in PyTorch.

We note that due to specific caching procedures of the Jax framework, each $x_{\mathrm{ehr}}$ instance has to be standardized into the same time-step length for the LSTM encoder to avoid out-of-memory issues. The Jax

framework requires that an LSTM encoder defines a static length of the sequences it is going to process, and then it caches this model in order to increase the training speed. If different sequence lengths are used, then Jax caches an instance of the LSTM encoder for each specific length to be used during each training cycle. The problem arises when dealing with a dataset that contains sequences of lengths that present high variance (i.e, many different sequence lengths for every data point in the dataset) In comparison, PyTorch does not use this approach and is able to process sequences of dynamic length with one single instance of the LSTM encoder. However, this comes at the cost of increased training times when comparing both frameworks. This problem was during the experimentation phase of our work, however we note that for the patient in-hospital mortality task the length of all $x_{\mathrm{ehr}}$ instances is standardized to 48hrs which does not generate this issue.

