# OpenReview forum: "Data-Driven Priors for Uncertainty-Aware Deterioration Risk Prediction with Multimodal Data"
_TMLR — Under review for TMLR_

### Review · Reviewer_fYZ9 · 2026-04-05

**Summary Of Contributions:**

This paper introduces MedCertAIn, a Bayesian uncertainty quantification framework for multimodal in-hospital mortality prediction that combines EHR time-series and chest X-ray images. The idea is to construct data-driven priors over neural network parameters by identifying "high-uncertainty" context samples through two strategies: modality-specific data corruptions (noise, crops, flips, etc.) and selection of training samples with low cross-modal similarity in a self-supervised (ConVIRT) latent space. These context points are used in a variational objective that regularizes the model to express high uncertainty on distributionally shifted inputs. The framework is evaluated on MIMIC-IV/MIMIC-CXR for in-hospital mortality prediction and compared against MedFuse, DrFuse, and an uninformative-prior BNN baseline.

Strengths:
- The problem (UQ in multimodal clinical settings) is well-motivated and understudied.
- The experimental protocol is thorough: 50 hyperparameter configs x 5 seeds, subpopulation analysis.
- Code is publicly available in JAX.

Weaknesses:
- The most directly related prior work ("Informative Priors Improve the Reliability of Multimodal Clinical Data Classification" - ML4H 2023) is not cited; the novelty delta is unclear.
- The theoretical framework is entirely from Rudner et al. (2024), reproduced "verbatim" per Appendix C.
- Standard UQ baselines (MC Dropout, Deep Ensembles) are absent; only one Bayesian baseline is compared.

**Additional Comments:**

- Section 3.1, Eq. 4: the label p-tilde is used but not defined until later in the appendix.
- Equation in Section 3.2.2: X_C^{ehr} appears on both sides of the last equation for the inter-intra selection (the right equation should be X_C^{cxr}).
- The paper cites Ovadia et al. twice as 2019a and 2019b. These are the same paper.

**Audience:**

Yes

**Audience Explanation:**

Uncertainty quantification for multimodal clinical data is an active and important research area. Researchers working on Bayesian deep learning, healthcare ML, and selective prediction would find the problem formulation relevant. The idea of constructing context sets from cross-modal disagreement is intuitive and potentially useful beyond the specific application studied here.

**Broader Impact Concerns:**

None. The paper appropriately targets a safety-critical domain (ICU mortality prediction) and the selective prediction mechanism. The use of publicly available MIMIC data does not raise additional privacy concerns.

**Claims And Evidence:**

No

**Claims Explanation:**

The central claim is that data-driven priors improve both predictive performance and uncertainty estimation in multimodal clinical settings. On standard metrics, the gains are modest but consistent: AUROC improves from 0.819 (MedFuse) and 0.820 (uninformative prior BNN) to 0.835, and AUPRC from 0.466/0.485 to 0.498 (Table 2). These differences appear reproducible across seeds given the tight standard errors. The selective prediction improvements are larger.

The selective prediction comparison is fundamentally unfair to the deterministic baselines. MedFuse and DrFuse were never designed for uncertainty-based deferral; they lack any uncertainty mechanism, so using Shannon entropy of their point predictions as a selection score is not a meaningful test of their selective prediction capability. The only fair UQ comparison is against the uninformative-prior BNN, where the gap is narrower (selective AUROC 0.711 -> 0.857, selective AUPRC 0.357 -> 0.599). The reported "195% improvement in selective AUPRC" (Section 5.1) is misleading because it is computed against a deterministic baseline that has no mechanism for selective prediction. The paper should report improvements relative to the uninformative BNN instead.

More importantly, standard UQ baselines are entirely absent. MC Dropout (Gal & Ghahramani, 2016) and Deep Ensembles (Lakshminarayanan et al., 2017) are the two most widely used UQ methods in healthcare ML, routinely included in UQ benchmarks (Ovadia et al., 2019; Nado et al., 2022, which are both cited by this paper). Their omission makes it impossible to contextualize MedCertAIn's improvements. A method claiming state-of-the-art UQ without comparing to ensembles or dropout cannot support that claim.

The ablation in Table 3 is informative but reveals that random corruptions alone achieve most of the performance gain , while the cross-modal similarity component adds only marginal improvements. This undermines the claim that the cross-modal similarity strategy is important.

Regarding novelty: the theoretical framework in Sections 3.1 and Appendix C is acknowledged as being from Rudner et al. (2024), with Appendix C.1 stating "We reproduce the derivation from Rudner et al. (2024) verbatim below." This is transparent, but it means the methodological novelty is limited to the context set construction (Section 3.2). Furthermore, a closely related prior work "Informative Priors Improve the Reliability of Multimodal Clinical Data Classification" (ML4H 2023) uses the same approach (data-driven priors with mean-field VI on MIMIC-IV + MIMIC-CXR) and is not cited anywhere in the paper. As it stands, the reader cannot determine what is new relative to that work.

Finally, evaluation is limited to a single clinical task (in-hospital mortality) on a single dataset pair with only 6,215 paired samples (4,485 training). The paper claims the framework is "scalable across diverse clinical settings" (Section 1) and "independent of the underlying fusion architecture" (Section 3), but neither claim is tested.

**Requested Changes:**

- The authors must formally cite and discuss "Informative Priors Improve the Reliability of Multimodal Clinical Data Classification", ML4H 2023. This prior work appears to employ a nearly identical methodology applied to the same MIMIC datasets. Given that the ablation studies in Table 3 indicate the majority of the performance gain is derived from random corruptions (which is the core contribution of the 2023 work), the authors must explicitly articulate the technical and empirical delta of MedCertAIn. A comparative discussion is required to justify the additional complexity of the ConVIRT-based cross-modal similarity selection.

- To support the claim of achieving state-of-the-art predictive performance and reliability, the authors must include standard UQ baselines, specifically Deep Ensembles and MC Dropout.

- The reporting of improvement percentages in the Abstract and Section 5.1 must be revised to ensure scientific accuracy. Currently, the '195% improvement' in selective AUPRC is calculated against a deterministic baseline (MedFuse) that lacks an inherent uncertainty mechanism.

- The manuscript repeatedly claims to provide 'calibrated predictions' and 'calibrated uncertainty estimates,' yet no formal calibration metrics are provided in the results. The authors must report Expected Calibration Error (ECE) and/or Brier Scores for all models in Table 2.

- I suggest the authors evaluate on at least one additional clinical task.

- The authors should provide a more detailed analysis of what the cross-modal similarity selection actually captures. Given that corruptions alone achieve most of the gain (Table 3), the authors should justify why the additional complexity of ConVIRT pre-training and similarity thresholding is worthwhile.

---

> ### Author Response · Authors · 2026-06-15
> **Response to reviewer 1/7**
>
> Thank you for the deep and constructive feedback.
>
> ### **Additional Uncertainty Baselines**
> We thank the reviewer for this suggestion. We agree that comparisons against standard uncertainty estimation methods are important for better positioning MedCertAIn relative to existing approaches.
> Following this recommendation, we have expanded the experimental evaluation to include several widely used uncertainty and calibration baselines: Deep Ensembles using 5 independently trained models, MC Dropout with 50 stochastic forward passes, and post-hoc temperature scaling. The results are shown in the following table (full mean ± std results will be reported in the paper.):
>
> | Model | AUROC | AUPRC | Selective AUROC | Selective AUPRC |
> |---|---:|---:|---:|---:|
> | MedFuse | 0.819 | 0.466 | 0.659 | 0.203 |
> | Temp. Scaling | 0.820 | 0.470 | 0.657 | 0.206 |
> | Deep Ensemble | 0.819 | 0.469 | 0.657 | 0.208 |
> | MC Dropout | 0.822 | 0.480 | 0.681 | 0.233 |
> | Uninformative Prior | 0.820 | 0.485 | 0.711 | 0.357 |
> | **MedCertAIn (Ours)** | **0.835** | **0.498** | **0.857** | **0.599** |
> | Improvement vs MedFuse (%) | +2.0% | +6.9% | +30.0% | +195.1% |
> | Improvement vs Uninformative Prior (%) | +1.8% | +2.7% | +20.5% | +67.8% |
>
> Across these added baselines, we observe that standard predictive performance, measured by AUROC remains largely comparable to vanilla MedFuse with a more significant difference in AUPRC. Among the standard uncertainty baselines, MC Dropout provides the strongest improvement in selective prediction. However, MedCertAIn consistently achieves stronger selective prediction performance, particularly in selective AUROC and selective AUPRC, demonstrating that its uncertainty estimates provide more effective ranking for deferral.

---

> ### Author Response · Authors · 2026-06-15
> **Response to reviewer 2/7**
>
> ### **Additional Clinical Tasks**
>
> We thank the reviewer for this suggestion. To broaden the empirical evaluation beyond the original endpoint, we expanded the evaluation to include multiple horizons: 48-hour, 3-month, 6-month, and 1-year mortality. These tasks preserve the same paired EHR–CXR multimodal input structure while testing whether MedCertAIn provides consistent benefits across clinically relevant prediction horizons.
>
> **3-month mortality**
> | Model | AUROC | AUPRC | Selective AUROC | Selective AUPRC |
> |---|---:|---:|---:|---:|
> | MedFuse | 0.656 | 0.174 | 0.582 | 0.078 |
> | Temp. Scaling | 0.656 | 0.174 | 0.582 | 0.078 |
> | Deep Ensemble | 0.657 | 0.180 | 0.583 | 0.077 |
> | MC Dropout | 0.656 | 0.175 | 0.556 | 0.102 |
> | Uninformative Prior | 0.651 | 0.146 | 0.581 | 0.079 |
> | **MedCertAIn (Ours)** | **0.660** | **0.183** | **0.683** | **0.196** |
> | Improvement vs MedFuse (%) | +0.6% | +5.2% | +17.4% | +151.3% |
> | Improvement vs Uninformative Prior (%) | +1.4% | +25.3% | +17.6% | +148.1% |
>
>
> **6-month mortality**
> | Model | AUROC | AUPRC | Selective AUROC | Selective AUPRC |
> |---|---:|---:|---:|---:|
> | MedFuse | 0.642 | 0.196 | 0.544 | 0.102 |
> | Temp. Scaling | 0.642 | 0.196 | 0.544 | 0.102 |
> | Deep Ensemble | 0.646 | **0.211** | 0.544 | 0.100 |
> | MC Dropout | 0.648 | 0.205 | 0.556 | 0.102 |
> | Uninformative Prior | 0.644 | 0.197 | 0.563 | 0.103 |
> | **MedCertAIn (Ours)** | **0.654** | 0.189 | **0.683** | **0.196** |
> | Improvement vs MedFuse (%) | +1.9% | -3.6% | +25.6% | +92.2% |
> | Improvement vs Uninformative Prior (%) | +1.6% | -4.1% | +21.3% | +90.3% |
>
> **1-year mortality**
> | Model | AUROC | AUPRC | Selective AUROC | Selective AUPRC |
> |---|---:|---:|---:|---:|
> | MedFuse | 0.636 | 0.227 | 0.545 | 0.137 |
> | Temp. Scaling | 0.636 | 0.227 | 0.545 | 0.137 |
> | Deep Ensemble | 0.638 | 0.230 | 0.544 | 0.136 |
> | MC Dropout | 0.637 | **0.231** | 0.545 | 0.135 |
> | Uninformative Prior | 0.593 | 0.197 | 0.537 | 0.167 |
> | **MedCertAIn (Ours)** | **0.639** | 0.222 | **0.661** | **0.252** |
> | Improvement vs MedFuse (%) | +0.5% | -2.2% | +21.3% | +83.9% |
> | Improvement vs Uninformative Prior (%) | +7.8% | +12.7% | +23.1% | +50.9% |
>
> Across these added tasks, MedCertAIn is consistently the strongest MedFuse-based variant in selective AUROC and selective AUPRC. The only exception in standard predictive performance is AUPRC for 6-month and 1-year mortality, where Deep Ensemble (6 month) and MC Dropout (1 year) variants perform better. However, MedCertAIn remains strongest on the selective prediction metrics.
>
> This supports one of the core intended behaviors of the method: examples assigned higher uncertainty are deferred, resulting in improved discrimination on the retained subset. This pattern is particularly important because, for many of the other baselines, selective prediction metrics remain below their corresponding standard AUROC/AUPRC values, indicating that their uncertainty or selection scores do not reliably identify cases whose removal improves performance.
>
> In contrast, MedCertAIn not only achieves the strongest selective AUROC and selective AUPRC across the evaluated mortality horizons, but also improves relative to its own standard predictive metrics under selective evaluation. This suggests that MedCertAIn’s uncertainty estimates provide a more effective ranking for deferral, which is the central objective of the selective prediction setting
>
> We also agree with the reviewer that our original wording was too broad regarding scalability across clinical settings and independence from the fusion architecture. The added deterioration horizons help us provide stronger evidence that MedCertAIn can be applied across related clinical prediction settings using the same paired EHR-CXR data.
> At the same time, we acknowledge that although the objective is formulated at the level of context-prior construction and loss regularization, and therefore does not require architectural changes to the fusion backbone, our experiments instantiate it using MedFuse. We have revised the manuscript to make this distinction explicit and no longer claim empirical architecture independence across multiple fusion models. Extending and validating MedCertAIn across additional fusion architectures remains an important direction for future work.

---

> ### Author Response · Authors · 2026-06-15
> **Response to reviewer 3/7**
>
> ### **Calibration Metrics**
> We have added ECE and Brier Score tables for all baseline models across the different mortality horizons evaluated: 48h, 3-month, 6-month, and 1-year mortality.
>
> **48-hour mortality**
> | Model | ECE | Brier |
> |---|---:|---:|
> | MedFuse | **3.215** | 0.102 |
> | Temp. Scaling | 3.527 | 0.102 |
> | Deep Ensemble | 3.455 | 0.102 |
> | MC Dropout | 3.613 | 0.102 |
> | Uninformative Prior | 4.625 | 0.102 |
> | **MedCertAIn (Ours)** | 4.323 | **0.100** |
>
> **3-month mortality**
> | Model | ECE | Brier |
> |---|---:|---:|
> | MedFuse | 0.979 | **0.068** |
> | Temp. Scaling | 1.545 | 0.069 |
> | Deep Ensemble | **0.847** | **0.068** |
> | MC Dropout | 0.937 | **0.068** |
> | Uninformative Prior | 1.313 | 0.070 |
> | **MedCertAIn (Ours)** | 1.492 | 0.076 |
>
> **6-month mortality**
> | Model | ECE | Brier |
> |---|---:|---:|
> | MedFuse | 1.168 | 0.092 |
> | Temp. Scaling | 0.831 | 0.092 |
> | Deep Ensemble | **0.699** | **0.091** |
> | MC Dropout | 1.452 | **0.091** |
> | Uninformative Prior | 2.065 | **0.091** |
> | **MedCertAIn (Ours)** | 2.105 | 0.092 |
>
> **1-year mortality**
> | Model | ECE | Brier |
> |---|---:|---:|
> | MedFuse | 2.086 | **0.118** |
> | Temp. Scaling | 2.007 | **0.118** |
> | Deep Ensemble | 2.718 | **0.118** |
> | MC Dropout | **1.879** | **0.118** |
> | Uninformative Prior | 5.447 | 0.135 |
> | **MedCertAIn (Ours)** | 2.399 | 0.119 |
>
> These results provide a clearer view of the relationship between calibration and selective prediction. MedCertAIn does not uniformly achieve the lowest ECE or Brier Score across all horizons; for example, it achieves the best Brier Score on 48h mortality, while other baselines perform better on some calibration metrics for the longer-term mortality tasks.
>
> We have revised the manuscript to reflect this distinction more precisely. Rather than framing MedCertAIn’s uncertainty estimates as uniformly better calibrated, we now describe the main benefit as improved predictive performance and stronger selective prediction through uncertainty-based ranking.
>
> This distinction is important because selective prediction evaluates whether uncertainty scores effectively prioritize and rank examples for deferral, whereas ECE and Brier Score evaluate absolute probability calibration.
> We also added this point to the limitations, noting that MedCertAIn does not consistently improve standard calibration metrics across mortality horizons and that calibration-aware training or post-hoc calibration remains a direction for future work.

---

> ### Author Response · Authors · 2026-06-15
> **Response to reviewer 4/7**
>
> ### **Revised Improvement Percentages for a Fair Comparison**
>
> We thank the reviewer for this comment. We agree that percentage improvements relative to deterministic baselines such as MedFuse and DrFuse can be misleading for selective prediction, since these models were not designed for uncertainty-based deferral.
>
> To ensure a fair interpretation of the gains, we have revised the manuscript to report comparisons relative to both MedFuse, as the original deterministic backbone, and the uninformative-prior BNN, as the most direct Bayesian baseline. We also avoid overstating percentage improvements against deterministic models and clearly distinguish reference comparisons from uncertainty-aware comparisons.
>
> We have revised the manuscript text in Section 5.1, and any related summary statements, to avoid emphasizing percentage improvements relative to deterministic models for selective prediction. Instead, we now report the main selective-prediction gains relative to the uninformative-prior BNN, which is the most direct uncertainty-aware baseline, while treating MedFuse as the deterministic backbone reference.

---

> ### Author Response · Authors · 2026-06-15
> **Response to reviewer 5/7**
>
> ### **Ablations and Role of Cross-Modal Similarity**
>
> We thank the reviewer for this important observation. We agree that Table 3 shows random corruptions account for the largest portion of the performance gain. We have revised the manuscript to make this explicit and to avoid overstating the marginal numerical contribution of the cross-modal similarity component.
>
> The role of the ConVIRT-based similarity strategy is not to replace random corruptions, but to provide a complementary, label-free mechanism for constructing informative context sets. Random corruptions are effective, but they require manually specifying perturbations that are appropriate for each modality and task. In contrast, the cross-modal similarity strategy uses a self-supervised CXR–EHR representation learned without clinical labels to identify samples with weak image–text alignment. These samples capture uncertainty arising from cross-modal disagreement, which is not directly modeled by modality-specific corruptions alone.
>
> We therefore clarify the technical delta of MedCertAIn as the combination of two sources of informative prior examples: (i) corruption-based samples that capture robustness to synthetic perturbations, and (ii) similarity-based patient samples that capture naturally occurring multimodal clinical inconsistencies in the latent space. While the corruption component contributes most of the gain in the current experiments, the similarity component provides a reusable and task-agnostic mechanism that does not require relabeling or redesigning the context set for each downstream endpoint.
>
> To further support this point, having added additional clinical evaluation tasks (i.e., 3-month, 6-month, and 1-year mortality), the ConVIRT-based context sets were reused without retraining or changing the similarity model, only the downstream supervised training objective was changed. This demonstrates that the self-supervised similarity component can serve as a reusable starting point for related clinical prediction tasks involving the same data modalities.
>
> We have revised the discussion to reflect this more precisely: the empirical gains from cross-modal similarity are modest in the current setting, but its value lies in providing a label-free, reusable, and clinically meaningful signal of multimodal disagreement. We also added this as future direction, noting that stronger task-adapted or clinically fine-tuned multimodal self supervised models may further improve the utility of similarity-based context selection.

---

> ### Author Response · Authors · 2026-06-15
> **Response to reviewer 6/7**
>
> ### **Relation to Prior Work**
>
> We thank the reviewer for raising this point. We have revised the manuscript to formally cite and discuss *Informative Priors Improve the Reliability of Multimodal Clinical Data Classification* (ML4H 2023), and to clarify how the present work extends that framework. However, we note that the cited preprint is a non-archival workshop paper.
>
> The present manuscript builds on this foundation in three main ways:
> 1. Whereas the ML4H 2023 work constructed context sets using synthetic corruption-based perturbations, MedCertAIn introduces an additional improvement using label-free, cross-modal self-supervised context selection strategy based on ConVIRT representations. This expands the context set beyond task-specific synthetic perturbations and allows the model to capture naturally occurring EHR–CXR disagreement as an additional source of uncertainty, without requiring clinical labels for context-set construction.
> 2. The present work provides a substantially larger empirical evaluation, spanning four mortality prediction endpoints: 48-hour, 3-month, 6-month, and 1-year mortality. It also compares against a broader set of deterministic and uncertainty-aware baselines, including MeTra, DrFuse, MedFuse, temperature scaling, MC Dropout, Deep Ensembles, and the uninformative-prior BNN.
> 3. The evaluation extends beyond predictive performance alone by reporting selective prediction, calibration metrics, subpopulation analyses, model ablations, and robustness across seeds and hyperparameters.
> We have revised the manuscript to make clear that MedCertAIn extends this previous informative-prior framework through label-free cross-modal context selection and a broader multimodal clinical evaluation setting.

---

> ### Author Response · Authors · 2026-06-15
> **Response to reviewer 7/7**
>
> ### **Additional Requested Changes**
>
> We thank the reviewer for carefully identifying notation and citation issues. The notation issues in Section 3.1 carried over from a previous manuscript version, Section 3.2.2 has also been reformated for clarity. We have updated the revised manuscript to use consistent notation throughout. These changes are purely notational and do not affect the method, implementation, training procedure, or reported results.
>
> We have also removed the duplicate Ovadia et al. citation and now cite the work consistently as a single 2019 reference.

---

### Review · Reviewer_CVjC · 2026-04-21

**Summary Of Contributions:**

The paper proposes MedCertAIn, a multimodal unncertainty-aware framework for clinical risk prediction that combines Bayesian neural networks with data-driven priors. It focuses on improving reliability in deployment by using uncertainty estimates to support selective prediction and deferral to help aid practitioners.

**Audience:**

Yes

**Audience Explanation:**

The ideas presented in the paper are not groundbreaking but the overall experiments are neat and the organization of the paper is of high quality.

One of the major issues I have, coming from the vision side, is the lack of comparisons to simple methods that are known to work well in practice, such as MC Dropout, Deep Ensembles, or even post-hoc calibration techniques. These approaches are often strong and competitive baselines for uncertainty estimation, and in many cases have the potential to outperform more complex Bayesian approximations. Without including such comparisons, it is difficult to assess whether the proposed method provides a meaningful advantage over established and widely used alternatives, particularly in terms of calibration and robustness. I'm looking forward to hearing why the authors decided not to include these benchmarks or whether these benchmarks are not feasible for multimodal scenarios, which I doubt -- I would imaging one can implement all three examples rather easily in multimodal settings.

**Claims And Evidence:**

Yes

**Claims Explanation:**

The paper has a clear practical motivation, focusing on deployment-relevant uncertainty through selective prediction and deferral rather than accuracy alone. I found the paper rather easy to read and informative. Overall, most of the claims in the paper are supported by experiments, and the language used throughout is appropriate.

The evaluations and ablations are generally well-structured. The multimodal setting is an emerging area, so there are fewer established benchmarks to compare against, and the paper does a reasonable job within these constraints. That said, the experimental scope could still be strengthened with additional baselines to better position the method relative to existing work.

**Requested Changes:**

As mentioned above, I would appreciate clarification from the authors regarding the absence of comparisons to simple and widely used uncertainty estimation methods such as MC Dropout, Deep Ensembles, and temperature scaling. These approaches are generally straightforward to implement and, I imagine that they can be adapted to multimodal settings with minimal effort, making them valuable baselines for benchmarking.

I appreciate the authors sharing the code but I doubt this code is of any use right now since their trainer is almost 3 thousand LOC with 75 imports. The authors mix and match way too many libraries, including jax as the main framework, TF as the backbone, but pytorch for data handling which makes this code unreadable. I kindly request a bare-bone implementation of the core evaluation method without training, so that others can easily benchmark it. For example, DrFuse [1] which the authors use as a benchmark has much cleaner implementation which allowed the authors to use it. They should show the same courtesy for researchers who most likely will follow-up on this work.

The method name is memorable, but it may lead to some confusion, as similar terms such as "MedCertain" or "Med-Certain" already appear in related contexts. This is not a major issue and does not require a change, but it may be worth considering for clarity and discoverability of the paper.

[1] https://github.com/dorothy-yao/drfuse

---

> ### Author Response · Authors · 2026-06-15
> **Response to reviewer 1/3**
>
> Thank you for the positive feedback and helpful suggestions.
>
> ### **Additional Uncertainty Baselines**
> We thank the reviewer for this suggestion. We agree that comparisons against standard uncertainty estimation methods are important for better positioning MedCertAIn relative to existing approaches.
> Following this recommendation, we have expanded the experimental evaluation to include several widely used uncertainty and calibration baselines: Deep Ensembles using 5 independently trained models, MC Dropout with 50 stochastic forward passes, and post-hoc temperature scaling. The results are shown in the following table (full mean ± std results will be reported in the paper.):
>
> | Model | AUROC | AUPRC | Selective AUROC | Selective AUPRC |
> |---|---:|---:|---:|---:|
> | MedFuse | 0.819 | 0.466 | 0.659 | 0.203 |
> | Temp. Scaling | 0.820 | 0.470 | 0.657 | 0.206 |
> | Deep Ensemble | 0.819 | 0.469 | 0.657 | 0.208 |
> | MC Dropout | 0.822 | 0.480 | 0.681 | 0.233 |
> | Uninformative Prior | 0.820 | 0.485 | 0.711 | 0.357 |
> | **MedCertAIn (Ours)** | **0.835** | **0.498** | **0.857** | **0.599** |
> | Improvement vs MedFuse (%) | +2.0% | +6.9% | +30.0% | +195.1% |
> | Improvement vs Uninformative Prior (%) | +1.8% | +2.7% | +20.5% | +67.8% |
>
> Across these added baselines, we observe that standard predictive performance, measured by AUROC remains largely comparable to vanilla MedFuse with a more significant difference in AUPRC. Among the standard uncertainty baselines, MC Dropout provides the strongest improvement in selective prediction. However, MedCertAIn consistently achieves stronger selective prediction performance, particularly in selective AUROC and selective AUPRC, demonstrating that its uncertainty estimates provide more effective ranking for deferral.

---

> ### Author Response · Authors · 2026-06-15
> **Response to reviewer 2/3**
>
> ### **Additional Clinical Tasks**
>
> We thank the reviewer for this suggestion. To broaden the empirical evaluation beyond the original endpoint, we expanded the evaluation to include multiple horizons: 48-hour, 3-month, 6-month, and 1-year mortality. These tasks preserve the same paired EHR–CXR multimodal input structure while testing whether MedCertAIn provides consistent benefits across clinically relevant prediction horizons.
>
> **3-month mortality**
> | Model | AUROC | AUPRC | Selective AUROC | Selective AUPRC |
> |---|---:|---:|---:|---:|
> | MedFuse | 0.656 | 0.174 | 0.582 | 0.078 |
> | Temp. Scaling | 0.656 | 0.174 | 0.582 | 0.078 |
> | Deep Ensemble | 0.657 | 0.180 | 0.583 | 0.077 |
> | MC Dropout | 0.656 | 0.175 | 0.556 | 0.102 |
> | Uninformative Prior | 0.651 | 0.146 | 0.581 | 0.079 |
> | **MedCertAIn (Ours)** | **0.660** | **0.183** | **0.683** | **0.196** |
> | Improvement vs MedFuse (%) | +0.6% | +5.2% | +17.4% | +151.3% |
> | Improvement vs Uninformative Prior (%) | +1.4% | +25.3% | +17.6% | +148.1% |
>
>
> **6-month mortality**
> | Model | AUROC | AUPRC | Selective AUROC | Selective AUPRC |
> |---|---:|---:|---:|---:|
> | MedFuse | 0.642 | 0.196 | 0.544 | 0.102 |
> | Temp. Scaling | 0.642 | 0.196 | 0.544 | 0.102 |
> | Deep Ensemble | 0.646 | **0.211** | 0.544 | 0.100 |
> | MC Dropout | 0.648 | 0.205 | 0.556 | 0.102 |
> | Uninformative Prior | 0.644 | 0.197 | 0.563 | 0.103 |
> | **MedCertAIn (Ours)** | **0.654** | 0.189 | **0.683** | **0.196** |
> | Improvement vs MedFuse (%) | +1.9% | -3.6% | +25.6% | +92.2% |
> | Improvement vs Uninformative Prior (%) | +1.6% | -4.1% | +21.3% | +90.3% |
>
> **1-year mortality**
> | Model | AUROC | AUPRC | Selective AUROC | Selective AUPRC |
> |---|---:|---:|---:|---:|
> | MedFuse | 0.636 | 0.227 | 0.545 | 0.137 |
> | Temp. Scaling | 0.636 | 0.227 | 0.545 | 0.137 |
> | Deep Ensemble | 0.638 | 0.230 | 0.544 | 0.136 |
> | MC Dropout | 0.637 | **0.231** | 0.545 | 0.135 |
> | Uninformative Prior | 0.593 | 0.197 | 0.537 | 0.167 |
> | **MedCertAIn (Ours)** | **0.639** | 0.222 | **0.661** | **0.252** |
> | Improvement vs MedFuse (%) | +0.5% | -2.2% | +21.3% | +83.9% |
> | Improvement vs Uninformative Prior (%) | +7.8% | +12.7% | +23.1% | +50.9% |
>
> Across these added tasks, MedCertAIn is consistently the strongest MedFuse-based variant in selective AUROC and selective AUPRC. The only exception in standard predictive performance is AUPRC for 6-month and 1-year mortality, where Deep Ensemble (6 month) and MC Dropout (1 year) variants perform better. However, MedCertAIn remains strongest on the selective prediction metrics.
>
> This supports one of the core intended behaviors of the method: examples assigned higher uncertainty are deferred, resulting in improved discrimination on the retained subset. This pattern is particularly important because, for many of the other baselines, selective prediction metrics remain below their corresponding standard AUROC/AUPRC values, indicating that their uncertainty or selection scores do not reliably identify cases whose removal improves performance.
>
> In contrast, MedCertAIn not only achieves the strongest selective AUROC and selective AUPRC across the evaluated mortality horizons, but also improves relative to its own standard predictive metrics under selective evaluation. This suggests that MedCertAIn’s uncertainty estimates provide a more effective ranking for deferral, which is the central objective of the selective prediction setting

---

> ### Author Response · Authors · 2026-06-15
> **Response to reviewer 3/3**
>
> ### **Code Quality and Reproducibility**
>
> We thank the reviewer for their thorough feedback and review of our code. We agree that the original codebase was difficult to navigate, particularly because most of the training, data loading, model initialization, checkpointing, and evaluation logic was concentrated in a single large trainer script. To address this, we have thoroughly reorganized the repository (i.e., [Anonymous Repository](https://anonymous.4open.science/r/medcertain_tmlr-8154/README.md)) to make it easier for external users to read, run, and adapt.
>
> - The previous monolithic trainer has been replaced by a much more straightforward implementation in which the main entry point is intentionally lightweight and the core functionality is clearly separated into modular components. In particular, data loading, model initialization, training, evaluation, checkpoint handling, constants, and metric computation have been moved into dedicated utility modules (utils/).
> - The revised codebase now clearly separates model training from evaluation. Users who want to reproduce the full pipeline can train the MedFuse baselines across seeds, initialize MedCertAIn from those checkpoints, and then fine-tune MedCertAIn using the provided scripts. Users who only want to benchmark the core evaluation protocol can instead load saved model outputs or checkpoints and run the evaluation path directly, without needing to rerun the full training pipeline.
> - We have also added a clearer documentation (README.md) describing the minimal required path configuration, checkpoint-loading behavior, and the scripts used for training and evaluation. Our goal with these changes is to make the repository substantially easier to use for external benchmarking while preserving the full experimental pipeline used in the paper.
>
> ### **Naming of “MedCertAIn”**
> We thank the reviewer for this suggestion. We acknowledge the potential for confusion with similarly named terms. Since the reviewer notes that this is not a major issue and does not require a name change, we retain the name MedCertAIn in the revised manuscript.

---

### Review · Reviewer_Ccgm · 2026-06-01

**Summary Of Contributions:**

This paper develops a Bayesian framework for uncertainty quantification in prediction tasks with multimodal healthcare data. The proposed method uses variational inference, and provide concrete ways to construct prior distributions. Uncertainty is quantified through the poseterior distributions, and the method is evaluated on MIMIC with electronic health records (EHR) and chest X-ray (CXR) images modals.

**Audience:**

Yes

**Audience Explanation:**

Uncertain quantification is of importance in safe deployment of prediction systems.

**Broader Impact Concerns:**

None.

**Claims And Evidence:**

No

**Claims Explanation:**

The main contribution of this paper is a new method for uncertainty quantification in prediction tasks. However, the interpretation of the proposed uncertainty measure remains somewhat unclear to me. For example, how should the output metric be interpreted, and what does this interpretation depend on, such as the model, parameters, or prior? In addition, what is the intuition/rationale behind the chosen thresholds? A discussion will be helpful.

**Requested Changes:**

1. As mentioned above, the interpretation of the proposed uncertainty measure is somewhat unclear. For example, how should the output metric be interpreted, and what does this interpretation depend on, such as the model, parameters, or prior? In addition, what is the intuition/rationale behind the chosen thresholds? A discussion will be helpful.
2. In Section 3.2.2, I am a bit confused by the selection criterion $\gamma_1 < t$: since $t = \gamma_1 - v\sigma$ and $v,\sigma>0$, isn't this condition always satisfied. (Similarly, for the $\gamma_4<t_4$).

---

> ### Author Response · Authors · 2026-06-15
> **Reponse to reviewer 1/2**
>
> We thank the reviewer for their feedback.
>
> **Shannon Entropy Score**
>
> In our experiments, uncertainty is measured as the Shannon entropy of the Monte Carlo predictive distribution induced by the variational posterior.
> This is a standard summary of predictive uncertainty: low entropy indicates that the predictive distribution concentrates most probability mass on one class, while high entropy indicates a more diffuse predictive distribution.
> For binary prediction, entropy is maximized when the predictive probability is 0.5, corresponding to maximal uncertainty between the two classes.
> This is also the principle used in our method: the uncertainty-aware prior encourages context-set examples to lie in high-entropy regions of the predictive distribution, so that the model learns to express uncertainty on samples selected to reflect corruption-based or cross-modal disagreement.
> The score is therefore input-dependent and reflects uncertainty in the learned Bayesian predictive distribution, which is influenced by the model architecture, learned variational parameters, and prior.
>
> **Rejection Thresholds**
>
> We note that the rejection thresholds used in selective prediction are evaluation thresholds rather than fixed clinical decision thresholds.
> We sweep thresholds across the rejection range to evaluate how well each method ranks predictions by uncertainty: a better uncertainty estimate should defer higher-uncertainty predictions earlier, leading to stronger selective AUROC and AUPRC.
> A deployment-specific threshold would depend on the acceptable risk and review capacity of the clinical setting.
>
> **Contributions**
>
> We would also like to emphasize that the claims of the paper are supported by both the results in the original submission and the expanded empirical evaluation added during the review period. In the original manuscript, MedCertAIn was evaluated on multimodal mortality prediction using predictive performance, calibration, selective prediction, ablations, and subpopulation analyses.
>
> In the revised manuscript, we further strengthen this evidence by adding additional mortality horizons, broader deterministic and uncertainty-aware baselines, and robustness analyses across seeds and hyperparameters. These results provide evidence that MedCertAIn is not only competitive in predictive performance, but also improves the reliability of uncertainty estimates for identifying predictions that should be deferred for further clinical review. We have revised the manuscript to present these findings more clearly while also acknowledging the limitations raised by the reviewers.

---

> ### Author Response · Authors · 2026-06-15
> **Response to reviewer 2/2**
>
> *In Section 3.2.2, I am a bit confused by the selection criterion*
> $\gamma_1 < t$ : since $t=\gamma_1-v\sigma$ and $v,\sigma > 0$ ,
> *isn't this condition always satisfied. (Similarly, for the* $\gamma_4 < t_4$)."
>
> We thank the reviewer for pointing this out. In the previous version, the notation for the selection thresholds was confusing because it did not clearly distinguish between sample-level similarity scores and the distribution-level statistics used to define the cutoffs. We have revised Section 3.2.2 to use consistent notation throughout. These changes are purely notational and do not affect the construction of the context sets, the method, training procedure, or reported results. We refer the reviewer to the updated PDF for revised formulation, and also include it below:
>
> ---
>
> **Inter-modal similarity.** For each training sample, we compute the cross-modal cosine similarity for each training sample and define the selection threshold $t_1$:
>
> $t_1 = \bar{\alpha} - v\cdot\sigma_\alpha$, $\qquad$ $\alpha_i =\cos(\Phi^{\textrm{ehr}}_i, \Phi^{\textrm{cxr}}_i)$
>
> where
>
> $\bar{\alpha} = \frac{1}{n}\sum_{i=1}^n \alpha_i$, $\qquad$ $\sigma_{\alpha} = \operatorname{std}\{\alpha_i\}_{i=1}^n$
>
> and $v\in[1,2].$
>
> We define the selected index set as $\mathcal{C}_1=$ { $i\in${$1,\dots,n$}: $\alpha_i < t_1$}, and the corresponding context sets:
>
> $X^{\textrm{ehr}}_{\mathcal{C}_1}$ = {$X^{\textrm{ehr}}_i : i\in\mathcal{C}_1$},
>
> $X^{\textrm{cxr}}_{\mathcal{C}_1}$ = {$X^{\textrm{cxr}}_i : i\in\mathcal{C}_1$}
>
>
> **Inter- and intra-modal similarity.** We additionally incorporate intra-modal similarity between each training sample and the mean latent representation vector for each modality:
>
> $\beta_i = \cos\left(\Phi^{\textrm{ehr}}_i, \overline{\Phi}^{\textrm{ehr}}\right),$ $\qquad$ $\gamma_i = \cos\left(\Phi^{\textrm{cxr}}_i, \overline{\Phi}^{\textrm{cxr}}\right)$
>
> where
>
> $\overline{\Phi}^{\textrm{ehr}} = \frac{1}{n}\sum_{i=1}^n \Phi_i^{\textrm{ehr}},$ $\qquad$ $\overline{\Phi}^{\textrm{cxr}} = \frac{1}{n}\sum_{i=1}^n \Phi_i^{\textrm{cxr}}.$
>
> We aggregate the similarities by simple averaging and define the selection threshold $t_2$:
>
> $t_2 = \bar{\delta} - v\sigma_{\delta},$ $\qquad$ $\delta_i = \frac{\alpha_i + \beta_i + \gamma_i}{3},$ $\qquad$ $\bar{\delta} = \frac{1}{n}\sum_{i=1}^n \delta_i,$ $\qquad$ $\sigma_{\delta} = \operatorname{std}\{\delta_i\}_{i=1}^n.$
>
> We define the selected index set as $\mathcal{C}_2=$ { $i\in${$1,\dots,n$}: $\delta_i < t_2$}, and the corresponding context sets:
>
> $X^{\textrm{ehr}}_{\mathcal{C}_2}$ = {$X^{\textrm{ehr}}_i : i\in\mathcal{C}_2$},
>
> $X^{\textrm{cxr}}_{\mathcal{C}_2}$ = {$X^{\textrm{cxr}}_i : i\in\mathcal{C}_2$}
>
> Finally, for training MedCertAIn, the high-uncertainty sample set is constructed by combining the corrupted training examples with either the inter-modal samples $\mathcal{C}_1$ or the inter- and intra-modal samples
> $\mathcal{C}_2$ selected using self-supervised latent-space similarity.

---

### Review · Reviewer_PmHL · 2026-06-02

**Summary Of Contributions:**

This paper proposes a predictive uncertainty framework, called MedCertAIn, which leverages multimodal clinical data for in-hospital risk prediction to improve model performance and reliability. It can provide Uncertainty Quantification (UQ) to highlight risky prediction which can be deferred to professional review. This work is necessary for the trustworthy implementation of healthcare AI.

Strengths:
1. MedCertAIn is meaningful for real application in healthcare AI;
2. The setting of multimodal clinical data empowers MedCertAIn to reach more accurate prediction;
3. The prediction of MedCertAIn has Shannon entropy score to reflect its uncertainty, which is a significant reference to decide whether professional reviews necessary.

Weeknesses:
1. The mathematical foundation is not clear(I will explain in the next session);

**Audience:**

Yes

**Audience Explanation:**

This topic is very crucial in healthcare AI. There exist a lot of work contributing on this area.

**Claims And Evidence:**

No

**Claims Explanation:**

1. I think the key math foundation in the paper is the variational objective in section 2.3:
$$min_{q_{\Theta}\in \mathcal{Q}} \mathbb{D}{KL}(q_{\Theta} || p_{\Theta|\mathcal{D}})\Longleftrightarrow max_{q_{\Theta}\in \mathcal{Q}}\mathcal{F}(q_{\Theta})$$


    Why are the expressions on the two sides equivalent?

    With $\mathcal{F}(q_{\Theta}) = \mathbb{E}{q_{\Theta}}[\log p(y_{\mathcal{D}} | x_{\mathcal{D}}, \Theta)] - \mathbb{D}{KL}(q_{\Theta} || p_{\Theta|\mathcal{D}})$

    Can you prove that $\mathbb{E}{q_{\Theta}}[\log p(y_{\mathcal{D}} | x_{\mathcal{D}}, \Theta)]$ is constant, or its increasing rate is slower than $\mathbb{D}{KL}(q_{\Theta} || p_{\Theta|\mathcal{D}})$?

    Instead, $max_{q_{\Theta}\in \mathcal{Q}}\mathcal{F}(q_{\Theta})$ cannot guarantee $min_{q_{\Theta}\in \mathcal{Q}} \mathbb{D}{KL}(q_{\Theta} || p_{\Theta|\mathcal{D}})$

2. In section 3.1, additional term called Uncertainty regularization $\mathbb{E}{q_{\Theta}}[\mathbb{E}{p{X_c, Y_c}}[log \tilde{p}(Y_c | X-c, \Theta)]]$ into $\mathcal{F}(q_{\Theta})$. What is $\tilde{p}$? a new distribution?

3. The paper clarifies that the larger of  $\mathbb{E}{q_{\Theta}}[\mathbb{E}{p{X_c, Y_c}}[log \tilde{p}(Y_c | X-c, \Theta)]]$, the bigger the uncertainty quantification of the context sample contains. How to link $\mathbb{E}{q_{\Theta}}[\mathbb{E}{p{X_c, Y_c}}[log \tilde{p}(Y_c | X-c, \Theta)]]$ to Shannon entropy score of equation (1).

By the way, why is Shannon entropy equation numbered as (1) while other equations above have no numbers?

**Requested Changes:**

My concern is mainly on math foundation.
The author can try to handle the listed problems above.

---

> ### Author Response · Authors · 2026-06-15
> **Response to reviewer**
>
> Thank you for these comments. We agree that these points are related, and we address them together here by referring to the full derivation provided in Appendix C.
>
> **Question 2.**
>
> Thank you for catching this notation error.
>
> The $\tilde{p}$ notation was carried over from a previous version of this manuscript.
> We changed it in favor of more precise notation.
> The updated notation is shown in the derivations in Appendix C.1, "A Family of Data-Driven Priors".
> We have updated the submission manuscript to show the correct notation in the main text.
> We would like to emphasize that this update in the manuscript is purely notational and does not change the method or the underlying implementation.
>
> **Question 1.**
>
> Thank you for your question.
> We provided a complete derivation of the variational objective in Appendix C.3, "Variational Inference with Uncertainty-Aware Priors". The formulation of the data-driven prior is described in detail in Appendix C.1 "A Family of Data-Driven Priors".
> Please let us know if you have any questions about the derivation.
>
> **Question 3.**
>
> Thank you for the question. We refer the reviewer to Appendix C.2, "Data-Driven, Uncertainty-Aware Priors for Fine-Tuning Pre-trained Models" where we provide the construction of the uncertainty-aware data-driven prior and the corresponding context distribution.
>
> We note that Shannon entropy is not directly optimized in the data-driven prior.
> The data-driven prior encourages neutral logits on context points, and neutral logits induce high-entropy predictive distributions.
> The Shannon entropy score is then used at evaluation time as the uncertainty measure for selective prediction.
>
> **Comment 1**
>
> Thank you for catching this formatting error. The equation was unintentionally numbered while the surrounding equations in Section 3 were unnumbered. We have corrected this in the updated manuscript.

---

### Author Response · Authors · 2026-06-15
**Summary of review and main contributions of the work**

We thank all reviewers for their constructive feedback.

All reviewers recognize the central contribution of MedCertAIn, which is the development of an uncertainty-aware Bayesian framework for multimodal clinical risk prediction, where informative context sets are used to guide the model toward reliable uncertainty estimates on samples likely to require clinical review. Our goal is to improve uncertainty-based ranking for selective prediction and deferral, which is central for safe clinical deployment, while maintaining or improving predictive performance.

In response to the reviewers’ suggestions, we substantially strengthened the empirical evaluation. We added the following:
1. **Additional standard uncertainty and calibration baselines**, including MC Dropout, Deep Ensembles, and temperature scaling,
2. **Additional clinical prediction tasks**:  3-month, 6-month, and 1-year mortality prediction. Across these added settings, MedCertAIn remains consistently strong in selective AUROC and selective AUPRC, supporting the claim that its uncertainty estimates are useful for identifying predictions that should be deferred for further review.
3. **Added formal calibration metrics**, including ECE and Brier Score, and revised the manuscript to distinguish calibration from selective prediction. These results clarify that MedCertAIn’s main advantage is not uniform improvement in probability calibration across all horizons, but improved uncertainty-based ranking for selective prediction.

We further clarified the relationship to prior work, including the earlier informative-prior framework, and made the technical delta of MedCertAIn explicit: combining corruption-based context construction with a label-free cross-modal similarity strategy that captures naturally occurring EHR--CXR disagreement. We also refined the ablation discussion to acknowledge that corruptions provide most of the current empirical gain, while the cross-modal strategy provides a reusable, task-agnostic mechanism for constructing clinically meaningful context sets.

Finally, we addressed mathematical and presentation concerns by aligning the main-text notation with the full derivation in Appendix C, removing the ambiguous (\tilde p) notation, and clarifying that the context regularization is implemented as an auxiliary cost term. These changes do not affect the implemented objective or experimental results, but make the mathematical presentation more consistent and transparent.

Overall, the revised manuscript provides a stronger empirical and methodological case for MedCertAIn as a practical framework for uncertainty-aware multimodal clinical prediction, particularly in settings where uncertainty estimates are used to support selective prediction and clinical deferral.